# Taxonomies for structuring models for World-Earth systems analysis of the Anthropocene: subsystems, their interactions and social-ecological feedback loops

Jonathan F. Donges[1,2], Wolfgang Lucht[1,3,4], Sarah E. Cornell[2], Jobst Heitzig[5], Wolfram Barfuss[1,6], Steven J. Lade[2,7,8], and Maja Schlüter[2]

[1]Earth System Analysis, Potsdam Institute for Climate Impact Research, Member of the Leibniz Association, Telegrafenberg A31, 14473 Potsdam, Germany
[2]Stockholm Resilience Centre, Stockholm University, Kräftriket 2B, 114 19 Stockholm, Sweden
[3]Department of Geography, Humboldt University, Unter den Linden 6, 10099 Berlin, Germany
[4]Integrative Research Institute on Transformations of Human-Environment Systems, Humboldt University, Unter den Linden 6, 10099 Berlin, Germany
[5]Complexity Science, Potsdam Institute for Climate Impact Research, Member of the Leibniz Association, Telegrafenberg A31, 14473 Potsdam, Germany
[6]Department of Physics, Humboldt University, Newtonstr. 15, 12489 Berlin, Germany
[7]Fenner School of Environment and Society, The Australian National University, Building 141, Linnaeus way, Canberra, Australian Capital Territory, 2601, Australia
[8]Bolin Centre for Climate Research, Stockholm University, Stockholm, Sweden

*Correspondence to:* Jonathan F. Donges (donges@pik-potsdam.de)

**Abstract.**

In the Anthropocene, the social dynamics of human societies have become critical to understanding planetary-scale Earth system dynamics. The conceptual foundations of Earth system modelling have externalised social processes in ways that now hinder progress in understanding Earth resilience and informing governance of global environmental change. New approaches to global modelling of the human World are needed to address these challenges. The current modelling landscape is highly diverse and heterogeneous, ranging from purely biophysical Earth System Models, to hybrid macro-economic Integrated Assessments Models, to a plethora of models of socio-cultural dynamics. World-Earth models capable of simulating complex and entangled human-Earth system processes of the Anthropocene are currently not available. They will need to draw on and selectively integrate elements from the diverse range of fields and approaches, so future World-Earth modellers require a structured approach to identify, classify, select, combine and critique model components from multiple modeling traditions. Here, we develop taxonomies for ordering the multitude of societal and biophysical subsystems and their interactions. We suggest three taxa for modelled subsystems: (i) biophysical, where dynamics is usually represented by "natural laws" of physics, chemistry or ecology (i.e., the usual components of Earth system models), (ii) socio-cultural, dominated by processes of human behaviour, decision making and collective social dynamics (e.g., politics, institutions, social networks, and even science itself), and (iii) socio-metabolic, dealing with the material interactions of social and biophysical subsystems (e.g., human bodies, natural resources and agriculture). We show how higher-order taxonomies can be derived for classifying and describing the interactions between two or more subsystems. This then allows us to highlight the kinds of social-ecological feedback loops where new

modelling efforts need to be directed. As an example, we apply the taxonomy to a stylised World-Earth system model that endogenises socially transmitted choice of discount rates in a greenhouse gas emissions game to illustrate the effects of social-ecological feedback loops that are usually not considered in current modelling efforts. The proposed taxonomy can contribute to guiding the design and operational development of more comprehensive World-Earth models for understanding Earth re-

silience and charting sustainability transitions within planetary boundaries and other future trajectories in the Anthropocene.

## 1   Introduction

### 1.1   Revisiting Earth system analysis for the Anthropocene

In the age of the Anthropocene, human societies have emerged as a planetary-scale geological force shaping the future trajectory of the whole Earth system (Crutzen, 2002; Steffen et al., 2007; Lewis and Maslin, 2015; Waters et al., 2016; Lenton and

Latour, 2018; Steffen et al., 2018). Cumulative greenhouse gas emissions and extensive modifications of the biosphere have accelerated since the neolithic and industrial revolutions, especially through the rapid globalisation of social-economic systems during the 20th century, threatening the stability of the interglacial state (Lenton et al., 2016) that has enabled the development and wellbeing of human societies (Rockström et al., 2009a; Steffen et al., 2015). Political and societal developments during the 21st century and their feedback interactions with the planetary climate and biophysical environment will be decisive for the

future trajectory of the Earth system  (Lenton and Latour, 2018; Steffen et al., 2018). Business-as-usual is taking the planet into a 'hothouse Earth' state unprecedented for millions of years in geological history (Winkelmann et al., 2015; Ganopolski et al., 2016), while calls for rapid decarbonisation of the global economic system to meet the Paris climate agreement (Rockström et al., 2017) will also have complex consequences involving an intensified entanglement of social, economic and biophysical processes and their resulting feedback dynamics, up to the planetary scale (Mengel et al., 2018). Despite extensive debate about

the Anthropocene (Lewis and Maslin, 2015; Hamilton, 2015; Brondizio et al., 2016; Zalasiewicz et al., 2017), and growing recognition of the limitations of current Earth system models for analysis and policy advice in the context of these shifting dynamics  (van Vuuren et al., 2012, 2016; Verburg et al., 2016; Donges et al., 2017a, b; Calvin and Bond-Lamberty, 2018), little has been done to address the fundamental challenge of systematically reviewing the conceptual foundations of Earth system modelling to include dynamic social processes, rather than externalising them (Bretherton et al., 1986, 1988).

To understand planetary-scale social-ecological dynamics, models of World-Earth systems are urgently needed (Schellnhuber, 1998, 1999; Rounsevell et al., 2014; van Vuuren et al., 2016; Verburg et al., 2016; Donges et al., 2017a, b, 2020; Calvin and Bond-Lamberty, 2018). Epistemologically, we conceptualise World-Earth systems as planetary-scale systems consisting of the interacting biophysical subsystems of the Earth, and the social, cultural, economic, and technological subsystems of the World of human societies. It should be noted here that in the context of global change analysis and modelling, the term 'Earth

system' was intended to include human societies and their activities and artefacts (Bretherton et al., 1988; Schellnhuber, 1998, 1999). However, in currently influential science and policy contexts, notably the Intergovernmental Panel on Climate Change (IPCC) (Flato, 2011; Flato et al., 2013), 'Earth system models' deal only with the physical dynamics of the atmosphere, ocean, land surface and cryosphere, and a limited set of interactions with the biosphere. While some might see tautology in the term

'World-Earth systems', we use it to highlight that human societies, their cultures, knowledge and artefacts (the 'World') should now be included on equal terms in a new family of models to conduct systematic global analyses of the Anthropocene. A fully co-evolutionary approach is needed, in the sense of representing social-ecological feedback dynamics across scales.

Future World-Earth modelling efforts will largely be pieced together from existing conceptualisations and modelling tools and traditions of social and biophysical subsystems, which encode the state of the art in our understanding of the Anthropocene. Current efforts in World-Earth systems modelling are highly stylised (e.g. Kellie-Smith and Cox (2011); Garrett (2015); Jarvis et al. (2015); Heck et al. (2016); Nitzbon et al. (2017); Strnad et al. (2019)), or tend to be proof-of-concept prototypes (Beckage et al., 2018; Donges et al., 2020). None operate yet in a process-detailed, well-validated and data-driven mode. To serve these nascent efforts in enabling World-Earth systems analysis of the Anthropocene, this article addresses the core question of which are the relevant categories within which World-Earth models, as essential scientific macroscopes (Schellnhuber, 1999), should operate. The problem for both scientific integration and real-world application is that the characteristic basis of the interactions of social and biophysical subsystems is often not explicit in current models. Often, the interactions between these subsystems are not recognised at all. By framing a taxonomy around the current dominant distinctions – and disciplinary divides – we can begin to explore links and feedback mechanisms between taxa in more structured, systematic and transdisciplinary ways. With this taxonomy, we develop initial tools and terminologies that enable model builders and model users to be clear about their social, cultural, epistemological and perhaps also axiological standpoints.

We want to emphasise that this taxonomic approach does not presuppose that there is "one world" (an ontological position) when models of different worlds are combined, nor do we intend it to serve as a universal blueprint for models of essentially everything. Instead, we argue that a taxonomy can help to focus modellers' attention better on the ontological and epistemic commitments within their models. This approach opens Earth system analysis to deeper dialogues with proponents of non-human actors as shapers of the world (Latour, 2017; Morton, 2013), or even the possibility of no world at all (Gabriel, 2013).

While the present article proposes a conceptual basis for World-Earth modelling, the proposed taxonomy is employed in the companion paper by Donges et al. (2020) to develop the operational World-Earth modelling framework copan:CORE. Here, this framework is cast into software and applied to construct and study an example of a novel World-Earth model that seeks to overcome the long-standing challenge of endogenising the choice of discount factors (describing how much societies value the present relative to the future) in climate mitigation studies.

## 1.2 Structuring the landscape of global environmental change models

Diverse scientific modelling communities aim to capture different aspects of social-ecological dynamics embedded in the Earth system up to planetary scales. Some processes operating in the Earth system are commonly described as being governed by the "natural laws" and generalizable principles of physics, chemistry and (to some extent at least) ecology (for example, atmosphere and ocean circulation as governed by the physical laws of fluid and thermodynamics), while others are thought to be dominated by human behaviour, decision making and collective social dynamics (e.g., the regularities underlying individual and social learning). This tendency for separate treatment of these different kinds of process in the natural and social sciences gives rise to problems when dealing with the many real-world subsystems that operate in both domains simultaneously. What

is more, different scientific communities use different methods and adhere to different viewpoints as to the nature and character of such subsystems and their interactions. There is now a number of conceptualisations of social-ecological or coupled human-environment systems in environmental, sustainability and Earth system science (e.g. Vernadsky (1929/1986); Schellnhuber (1998); Fischer-Kowalski and Erb (2006); Jentoft et al. (2007); Biggs et al. (2012)) but we see a pressing need to structure modelling efforts across communities, providing a joint framework while maintaining the conceptual flexibility required for successful cross-disciplinary collaboration.

Here, we propose a taxonomic framework for structuring the multitude of subsystems that are represented in current mathematical and computer simulation models. The motivation for proposing such an ordering scheme is:

1. to provide the means for collecting and structuring information on what components of social-ecological systems relevant to global change challenges are already present in models in different disciplines,

2. to point out uncharted terrain in the Earth system modelling landscape, and

3. to provide the foundations for a systematic approach to constructing future co-evolutionary World-Earth models, where feedback mechanisms between components can be traced and studied. This conceptual work aims to contribute to a central quest of sustainability science (Mooney et al., 2013) that "seeks to understand the fundamental character of interactions between nature and society." (Kates et al., 2001).

## 1.3 Definitions and explanations of key terms

In this article, we use the term subsystem to refer to any dynamic component in models of World-Earth systems. In this broad category, we can include both the kinds of subsystems that are governed mainly by "natural laws" of physics, chemistry or ecology (e.g., seasonal precipitation, ocean nutrient upwelling) and those that are governed mainly by human behaviour, decision making and collective social dynamics (e.g., international food trade, carbon taxes). Many scientific communities similarly make this distinction between biophysical ("natural", ecological, environmental) subsystems and socio-cultural (social, human, "anthroposphere") subsystems. We also highlight socio-metabolic subsystems at the overlap of societal and natural "spheres" of the Earth system (Fig. 1). We suggest that explicit attention to these subsystems and their interactions is needed in order to deepen the understanding of transformative change in the planetary social-ecological system, making a valuable contribution to the design and operational development of future, more comprehensive World-Earth models for charting sustainability transitions into a safe and just operating space for humanity (Rockström et al., 2009a; Raworth, 2012; Dearing et al., 2014).

A further note on the term *biophysical*: here, we use this word as a shorthand term to refer to Earth's interacting living and non-living components, encompassing geophysical (climatic, tectonic, etc.), biogeophysical, biogeochemical and ecological processes. These categories are significant in Earth system science because feedbacks involving these processes tend to have different dynamic characteristics. Accordingly, they have been dealt with very differently in Earth system analysis and modelling (Charney et al., 1977; Gregory et al., 2009; Stocker et al., 2013).

The co-evolution of Earth's geosphere and biosphere is a central concept in Earth system science (Lovelock and Margulis, 1974; Budyko et al., 1987; Lovelock, 1989; Schneider et al., 2004; Lenton et al., 2004; Watson, 2008), but the global models

that currently dominate the field represent just a snapshot of the system, focused on the biophysical dynamics that play out over decades to centuries. We use the term co-evolution to describe the complex dynamics that arise from the reciprocal interactions of subsystems, each of which changes the conditions for the future time evolution of the other (not excluding, but not limited to processes of Darwinian co-evolution involving natural selection). Earth system models (ESMs) include key physical feedbacks, and increasingly permit the investigation of biophysical feedbacks, but as we have indicated, they lack socio-metabolic and socio-cultural subsystems, relying on narrative-based inputs for dealing with anthropogenic changes. Integrated assessment models (IAMs) used in the global change context (Edenhofer et al., 2014; van Vuuren et al., 2016) include some interactions of social and biophysical subsystems in order, say, to assess potential economic consequences of climate change and alternative climate policy responses. But they lack the kinds of interactions and feedback mechanisms (e.g., by impacts of climatic changes on socio-metabolic subsystems, or by the effects of socio-cultural formation of public opinion and coalitions in political negotiations on environmental policies) that societies throughout history have shown to be important which is revealed, e.g., by studies of social-ecological collapse and its connection to past climate changes (Weiss and Bradley, 2001; Ostrom, 2009; Donges et al., 2015; Cumming and Peterson, 2017; Barfuss et al., 2020). To explore and illustrate the consequences of these typically neglected interactions and feedbacks, we have studied a conceptual model that gives rise to complex co-evolutionary dynamics and bifurcations between qualitatively different system dynamics: a model of socially transmitted discount rates in a greenhouse gas emissions game, discussed in Section 4.

For completeness, we also provide brief definitions of our working terminology: a "link" or "interaction" is a causal influence of one subsystem on another that is operationally non-decomposable into smaller links; a "mechanism" is a micro-description of how exactly this causal influence is exerted; a "process" is a set of links that "belong together" from some suitable theoretical point of view; a "loop" is a closed path in the network of links; and an "impact" of a link is the change in the target system attributable to this link.

We should note here that this taxonomy is dealing with causal narratives from different scientific disciplines that are encoded in models, and as such, it does not require any a priori theories and hypotheses about causality. Causal narratives are our starting point because they are necessary for and are explicitly encoded in simulation modelling - and our classification lets us interrogate them more systematically and exposes them explicitly.

## 2 A taxonomy of subsystems in World-Earth systems models

In this section, we introduce the biophysical (ENV), socio-metabolic (MET), and socio-cultural (CUL) taxa for classifying subsystems in models of World-Earth systems (Fig. 1). For each taxon, we give examples of corresponding subsystems from different modelling fields. We also discuss how the suggested taxonomy relates to earlier conceptualisations of human societies embedded in and interacting with environmental systems (Sect. 2.4).

We have followed three guidelines in constructing this taxonomy for models of World-Earth systems:

1. *Compactness*, because we aim at a "top-level" framework that is useful and tangible, with as few classifications as possible, covering the scope of co-evolutionary modelling research parsimoniously and in a self-containing way.

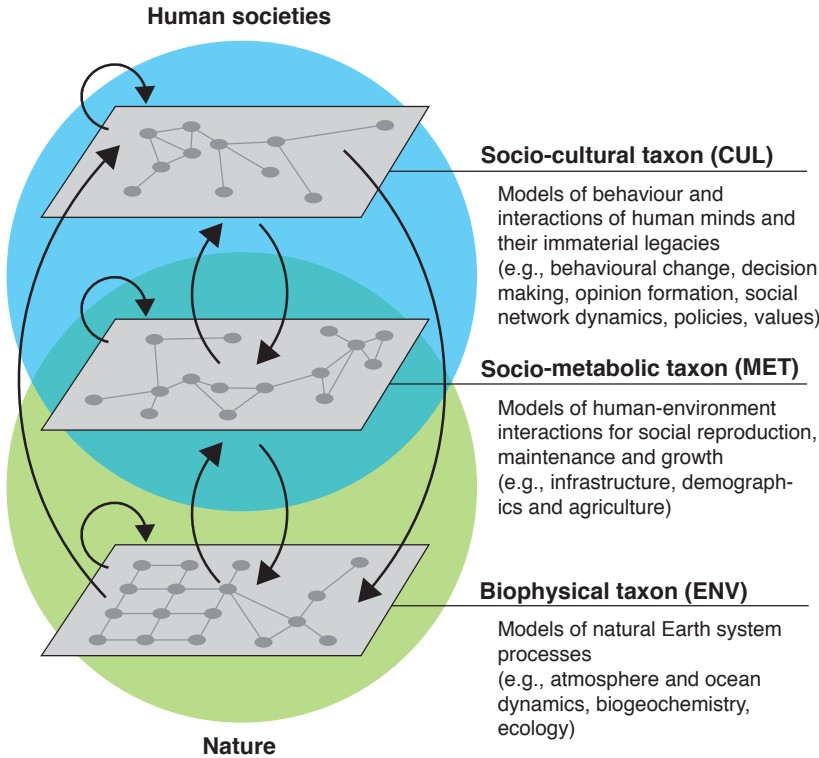

**Human societies**

**Socio-cultural taxon (CUL)**

Models of behaviour and interactions of human minds and their immaterial legacies (e.g., behavioural change, decision making, opinion formation, social network dynamics, policies, values)

**Socio-metabolic taxon (MET)**

Models of human-environment interactions for social reproduction, maintenance and growth (e.g., infrastructure, demograph-ics and agriculture)

**Biophysical taxon (ENV)**

Models of natural Earth system processes (e.g., atmosphere and ocean dynamics, biogeochemistry, ecology)

**Nature**

**Figure 1.** Proposed taxonomy of subsystems in World-Earth systems models. The blue and green overlapping discs represent the current discipline-based domains in which the subsystems and processes of nature, human societies, and their interactions are modelled. Our scheme structures this continuum into three taxa (light grey layers) for model subsystems (dark grey discs): (i) a biophysical taxon (ENV), (ii) a socio-metabolic taxon (MET), and a socio-cultural taxon (CUL). Links within and between these modelled subsystems (shown as black arrows in the figure) can further be classified using a $3 \times 3$ taxonomy of interactions (Fig. 2, Sect. 3).

2. *Compatibility* with existing disciplines and research fields within, between and beyond the persistent natural/social sciences divide, because we view the scientific endeavour of understanding links and feedbacks in co-evolutionary World-Earth systems as an integrative and transdisciplinary opportunity.

3. *Operative capacity* for model classification and construction, because we want to advance efforts rapidly in World-Earth modelling. This guideline differs from the previous two in that it deals with practical aspects of modeling. We include it because it flags the need for critical reflection on the suitability of combined models for the tasks at hand. We want to be able to expand the scope of modelling to be more inclusive, allowing more differentiation and well-founded permutations of approaches.

Models encode knowledge outside of the mind of the modeller, so these guiding principles are intended to ensure that bridging across currently very distinct modelling approaches still permits tracing back how the techniques relate to the theories, assumptions, and framings of the contributory disciplines.

The proposed taxonomy reflects the longstanding structure – and the underlying divides – of the scientific disciplines dealing with the respective subsystems. We argue that it also provides a blueprint for navigating the fragmented modelling landscape and bringing new opportunities for cross-disciplinary bridging. The anthropocentric and dialectic distinction between the realms of nature or "the environment" and of human societies has a long intellectual history. Deep philosophical and scientific puzzles are connected with the attempts to draw a sharp distinction between these domains, and to satisfactorily integrate properties such as mental states, intentions, and life itself.

With the progressive improvements in biophysical Earth system modelling (Reichler and Kim, 2008; Steffen et al., 2020) and the concomitantly growing reliance on model-based insights for global decision-making over a wider range of urgent sustainability issue (National Research Council, 2007; Rounsevell et al., 2014; Calder et al., 2018), as is the case for example for the Paris climate agreement (UNFCCC, 2015) informed by the IPCC (Stocker et al., 2013; Barros et al., 2014; Edenhofer et al., 2014) and the policy processes derived from it, these conceptually challenging issues can now have direct practical implications. Illustration such different conceptions of Earth system processes, in models of the contemporary Earth system, land vegetation can be treated as inanimate carbon, a transpiration "pump" affecting precipitation and soil moisture patterns (e.g. Sitch et al. (2003)), or as the animate matter of biodiverse ecosystems that sustain human communities (e.g. (Purves et al., 2013)). Similarly, different assumptions in models about non-material factors such as human rationality, cognition, motivations, institutions and social connections lead to very different likelihoods for alternative sustainability pathways for the world's economies and material resource use  (Donges et al., 2017b; Müller-Hansen et al., 2017; Beckage et al., 2018; Otto et al., 2020b).

For these reasons, we follow a pragmatic approach in proposing a taxonomic framework that draws upon examples and allows for overlap between the domains of nature and human societies, where materiality meets intention (noting that in complex social-ecological systems, purposeful intervention will be accompanied by unintended or unanticipated side effects). Following this approach, modelled subsystems in the biophysical taxon are situated in the material domain of nature, those in the socio-metabolic taxon lie in the overlap domain, and those in the socio-cultural taxon reside in the immaterial domain of human cultures (Fig. 1).

## 2.1 Biophysical taxon

The biophysical taxon (ENV) contains the processes and subsystems that are typically included in current comprehensive Earth system models, but views them from the perspective of the Anthropocene shift to human "co-control". These subsystem models are governed by deterministic and stochastic mathematical equations, often developed from first principles about the physical relationships involved. There is a case for subdividing the biophysical taxon into an ecological subtaxon (subsystems associated with life) and a geophysical subtaxon (subsystems not associated with life), since they have distinct, albeit co-evolving dynamics (Vernadsky, 1929/1986; Lenton et al., 2004), and this subdivision would correspond to widely accepted

geosphere/biosphere conceptualisations of the Earth system (Bretherton et al., 1986, 1988; Seitzinger et al., 2015). However, we apply our principle of compactness, because geosphere-biosphere links and processes have been comprehensively documented over the past few decades, as they underpin current Earth system and global integrated assessment modelling. Rather than retracing these links (after all, the existing models are not going to be completely reconfigured in light of the issues we explore in this paper), we have opted to take today's state of the art in biophysical global modelling as our main point of departure, following the principle of compatibility introduced above.

Earth system models have developed from coupled atmosphere-ocean general circulation models, progressively coupling in components describing biogeochemical and biogeophysical dynamics. On decade-to-millennium time scales relevant for the analysis of anthropogenic climate change and its medium-term consequences, examples of these modelled subsystems where human-controlled dynamics are prominent concerns include atmospheric chemistry, ocean productivity, sea ice, land vegetation, and major elemental cycles such as those of nitrogen, phosphorus, and sulfur (Bretherton et al., 1986, 1988). Furthermore, as it becomes clearer that palaeoclimate models can play a vital role in "deep future" studies of human-controlled processes in the Anthropocene, Earth system dynamics operating on longer time-scales are relevant (Zeebe and Zachos, 2013; Steffen et al., 2018). So for these purposes, the biophysical taxon would include subsystems involving the lithosphere (e.g., rock weathering, isostatic depression and rebound associated with the advance and retreat of ice sheets on land) and even external drivers such as large-body impacts (Brugger et al., 2017), if these provide "natural experiments" or analogues for future change.

Research fields dealing with models of subsystems belonging to the biophysical taxon include, among others, geophysics, meteorology, oceanography, biology, ecology, biogeochemistry, and geology. Few of these sciences have yet grasped the methodological and theoretical tools for dealing with the human dimensions of anthropogenic change. From our planetary-scale perspective, the ENV taxon exhibits a substantial overlap with categories such as models of "the environment", "nature" or "ecology", with their specific disciplinary connotations, although many of these models have tended to be small-scale, context-specific and idiographic. An exception from this are global dynamic vegetation models such as LPJ (Sitch et al., 2003), which focus, however, on representing the physical dynamics of ecological processes and structures in an Earth system context and not on ecological dynamics as such (i.e., interactions between living organisms). We note a current drive for further refinements of ecological dynamic network processes in large-scale modelling (Purves et al., 2013; Harfoot et al., 2014) within the ENV taxon that may improve global-scale conceptualisations of ecosystems in ways compatible with both Earth system modelling and socio-ecological systems research and resilience thinking.

## 2.2 Socio-metabolic taxon

The socio-metabolic taxon contains processes and subsystems that form the material basis and products of societies, making direct interconnections between human societies and the biophysical environment that sustains them. This taxon comprises models of demographics and social structure (e.g., population size, age/sex distribution, health parameters; and social categories with material or resource-use consequences, such as class, clan, caste, ethnicity). It also includes "the technosphere": society's artefacts, factors of production and technologies (e.g. labour, land, capital, natural resources, raw material, energy;

tools, machines, infrastructure; cultivated landscapes, domesticated animals and plants), and economic systems (manufacturing, distribution and consumption of goods and services) (Haff, 2012, 2014; Mooney et al., 2013; Herrmann-Pillath, 2018).

The broad field of economics currently dominates descriptions of parts of the socio-metabolic taxon in quantitative models, but many other disciplines such as geography, industrial metabolism, social ecology, and science and technology studies also play a role. In modelling terms, this taxon typically involves representations of both the biophysical planet Earth and the socio-cultural World of human societies. This implies hybrid models of the type that are currently included in Integrated Assessment Models of global change, and entails strong simplifying assumptions. We suggest that our approach can bring much-needed clarity and transparency about the role of such models in understanding World-Earth systems (c.f. van Vuuren et al. (2016)). One should note that IAMs and economic models are typically expressed in terms of financial value and not material flows that directly interact with subsystems in ENV (mostly empirical input-output theories of economics being an exception, Leontief (1936)).

## 2.3 Socio-cultural taxon

The socio-cultural taxon contains processes and subsystems that are described in models of the behaviour of human minds and their immaterial legacies, abstracted from their biophysical foundations and often described as lying in the realm of human agency (Otto et al., 2020b). Of the three taxa proposed, processes and subsystems in the socio-cultural taxon are the least formalised in mathematical and computer simulation models so far, despite substantial efforts in this direction in many fields of the social sciences (e.g. Farmer and Foley (2009)) and a likelihood that they may be only partly formalizable. Research fields dealing with models of processes and subsystems in the socio-cultural taxon include sociology, anthropology, behavioural economics, political science and social ecology. Our taxonomic approach can enable the diverse modelling activities now underway to engage more directly with the incipient World-Earth modelling effort.

Examples of modelled subsystems in this taxon include individual and collective opinions, behaviours, preferences, and expectations and their social network dynamics; information and communication networks; institutions and organisations; financial markets and trade; political processes; social norms and value systems (Mooney et al., 2013). Notably, the CUL taxon can also include processes of digital transformation and artificial intelligence that increasingly restructure and shape the socio-cultural sphere of human societies. It also provides a locus for debating the challenge of reflexiveness in science, especially in fields where modelling plays a vital role in shaping knowledge and action (Yearworth and Cornell, 2016). For instance, future World-Earth modelling will have to grapple with ways to recognize Earth system science as an endogenous generator of scientific conceptions of 'Earth'. Relevant for modelling efforts, socio-cultural subsystems can vary on substantially different time scales. Near instantaneous information exchanges are possible on online social networks and within and between increasingly advanced algorithms (e.g. algorithmic trading systems on financial markets), while elections and governance processes act on the order of years. Formal institutions (e.g. laws) change on the order of decades and informal institutions (e.g. religions) develop over time frames on the order of centuries to millennia (Williamson, 1998; Otto et al., 2020a).

## 2.4 Relations to other conceptualisations of social-ecological systems

Our model-centred taxonomy is inspired by previous systemic conceptualisations of human societies embedded in the Earth system, building upon them in a way that may help to bridge across diverse disciplines and theoretic traditions.

In one of the earliest Earth system conceptualisations, Vernadsky (1929/1986) distinguishes the inanimate matter of the geosphere, the living biosphere, and the noosphere of networked consciousness, the latter reverberating in recent conceptualisations of the technosphere and planetary human-Earth system interactions (Herrmann-Pillath, 2018; Lenton and Latour, 2018). Along these lines, Schellnhuber (1998, Fig. 34) introduced the ecosphere (directly corresponding to our ENV taxon, entailing geophysical and ecological interactions), the anthroposphere (broadly related to MET, but with some socio-cultural features), and the global subject (closely related to CUL).

Conceptualisations in resilience theory, ecological economics and sustainability science emphasise the interactions and interdependence of biosphere and society (Brundtland, 1987; Folke, 2006; Folke et al., 2011), with many sustainability practitioners adding the economy to make "three pillars" or a "pie of sustainability" consisting of economy embedded in society embedded in biosphere (Folke et al., 2016). These fields have typically focused on local to regional geographic scales or specific sectors, and have not placed much emphasis on global modelling, but in general terms, their view of society contains aspects of our MET taxon, while "the economy" is more restricted than MET. Herrmann-Pillath (2020) argues that the field of ecological economics would benefit from more attention to the creative processes of 'art', which we would frame as CUL aspects that are largely absent from current conceptualisations in that field and also more broadly (as also argued by Jax et al. (2013); Woroniecki et al. (2020)).

Fischer-Kowalski and Erb (2006) explicitly develop the concept of social metabolism, in terms of the set of flows between nature and culture, in order to describe deliberate global sustainability transitions. Governance-centred classification schemes in social-ecological systems research (Jentoft et al., 2007; Biggs et al., 2012), in the tradition of Ostrom (Ostrom, 2009), can also be brought into our taxonomy. Categories of the governance (sub)system link CUL and MET, and the (sub)system to be governed (ENV and MET) links the biophysical resources to be used with the social agents who will use them.

The taxonomy approach means that things that were previously included in models as opaque and unquestioned systems can be unpacked and critically examined. This would be of particular benefit to model users who were not the model builders. For example, education may be explicitly linked to demography (as in various integrated assessment models), so typically would be treated as a quantifiable and accumulable process in the MET taxon: i.e., investment in women's education results in a lower birth rate and therefore less future land use. In CUL, education would perhaps be treated in a more relational way - dealing with the spread of ideas, development of communities, changes in power structures etc.

## 3 Taxonomy of subsystem interactions in World-Earth systems models

In this section, we describe a taxonomy of modelled interactions between subsystems that builds upon the taxonomy of subsystems. The three taxonomic classes for World-Earth subsystems give rise to nine taxa for directed interactions connecting these subsystems. Given a pair of taxonomic classes of subsystems $A$ and $B$, the taxonomic class for directed interactions

between $A$ and $B$ is denoted as $A \rightarrow B$. Here, a directed interaction is understood in the sense of a modelled subsystem in $A$ exerting a causal influence on another modelled subsystem in $B$. For example, greenhouse gas emissions produced by an industrial subsystem in MET that exert an influence on the Earth's radiative budget in ENV would belong to the interaction taxon MET $\rightarrow$ ENV. Three of the nine interaction taxa correspond to self-interactions within taxa, while six interaction taxa connect distinct subsystem taxa (Fig. 2).

In the following, we focus on describing examples of such modelled interactions between pairs of subsystems that are potentially relevant for future trajectories of World-Earth systems in the Anthropocene and give examples of published models containing them. The content presented in the subsections necessarily differs in scope and depth reflecting today's dominant modelling priorities, but we have aimed to ensure the information is comparable. All subsections below provide (i) a general description of the interaction taxa with some examples, and (ii) a summary of how these interactions are represented in current models.

Furthermore, possible extensions of our taxonomic approach to classify feedback loops and more complex interaction networks between subsystems are discussed (Sect. 3.10). We acknowledge that finding a conceptualisation that is satisfactory for all purposes is unlikely, but our particular pragmatic taxonomy can be useful for constructing models of World-Earth systems. It has already proven fruitful in the development of the copan:CORE open World-Earth modelling framework (Donges et al., 2020) by guiding the choice of process classes and entities that can be described in the framework as well by defining the coupling interfaces of model components that can be integrated using copan:CORE.

## 3.1 ENV → ENV: Biophysical Earth system self-interactions

This taxon encompasses interactions between biophysical subsystems of the type studied in current process-detailed Earth system models such as those in the CMIP5 model ensemble (Taylor et al., 2012) used in the IPCC reports (Stocker et al., 2013). For example, this includes modelled geophysical fluxes of energy and momentum between atmosphere and ocean, interactions between land vegetation, atmospheric dynamics and the hydrological cycle, or, more generally, exchanges of organic compounds between different compartments of biogeochemical cycles (excluding human activities here).

A detailed representation of these biophysical interactions is largely missing so far in current first attempts at modelling social-ecological dynamics at the planetary scale (e.g. Kellie-Smith and Cox (2011); Heck et al. (2016)). However, emerging socio-hydrological (Di Baldassarre et al., 2017; Keys and Wang-Erlandsson, 2017) and agent-based land-use dynamics models at regional scales (Arneth et al., 2014; Rounsevell et al., 2014; Robinson et al., 2017) include some processes involving interactions between biophysical subsystems such as the atmosphere, hydrological cycles and land vegetation.

## 3.2 ENV → MET: Climate impacts, provisioning and regulating ecosystem services, etc.

This taxon describes modelled interactions through which biophysical subsystems exert an influence on socio-metabolic subsystems. Relevant examples in the context of global change in the Anthropocene include the impacts of climate change on human societies (Barros et al., 2014) such as damages to settlements, production sites and infrastructures and supply chains (Otto

|  | CUL | MET | ENV |
|---|---|---|---|
| **CUL** | **CUL→ CUL:** social networking, individual and social learning, behavioural and value changes, institutional and policy dynamics | **CUL→ MET:** socio-economic governance, demand, value-driven consumption, expressions of culture in required infrastructure | **CUL→ ENV:** environmental governance, nature conservation areas, cultural landscapes, parks, sacred places |
| **MET** | **MET→ CUL:** needs, constraints, supply of valued goods, effects of technological innovations, monitoring, observation | **MET→ MET:** interlinkage of systems of infrastructure, supply chains, demographic change, agriculture, material economics | **MET→ ENV:** Greenhouse gas emissions, land-use change, extraction of resources, chemical pollution and wastes, footprints |
| **ENV** | **ENV→ CUL:** Environmental embedding and foundations of culture, observation, monitoring, cultural ecosystem services | **ENV→ MET:** Climate impacts, resource flows, provisioning and regulating ecosystem services | **ENV→ ENV:** atmosphere-ocean-land couplings, geophysics, biogeochemistry, ecological networks, supporting ecosystem services |

**Figure 2.** Taxonomic matrix for classifying directed interactions between subsystems in World-Earth systems models. This $3\times3$ classification system builds upon the taxonomy of three classes for subsystems introduced in Sect. 2. The unshaded matrix elements (here containing examples of interactions) correspond to the interaction arrows drawn between the three subsystem taxa shown in Fig. 1. Shaded elements correspond to self-interactions. The examples for directed interaction mechanisms given in the matrix elements are indicative and based on our particular areas of research.

et al., 2017), impacts on agriculture or human health, but also provisioning and regulating ecosystem services such as resource flows (Millennium Ecosystem Assessment, 2005).

Some of these interactions such as climate change impacts are now being included in IAMs (a prominent example being the DICE model, Nordhaus (1992)) and stylised models (for example Kellie-Smith and Cox (2011) and Sect. 4), but there remain challenges, e.g. in estimating damage functions and the social cost of carbon (Nordhaus, 2017). Influence from weather and climate on agriculture are studied on a global scale using model chains involving terrestrial vegetation models such as LPJ (Sitch et al., 2003) and agricultural economics models such as MAgPIE (Nelson et al., 2014). As another example, models of the distribution of vector-born diseases such as Malaria are employed to assess the impacts of climate change on human health (Caminade et al., 2014).

## 3.3 ENV → CUL: observation, monitoring, cultural ecosystem services, etc.

This taxon contains modelled interactions through which the state of the biophysical environment directly influences socio-cultural subsystems. These links can be mediated through the observation, monitoring and assessment of environmental change from local to global scales (e.g., chemical pollution, deforestation or rising greenhouse gas concentrations in the atmosphere) by social actors that in turn are processed by public opinion formation and policy making in socio-cultural subsystems (Mooney et al., 2013). The links described by the ENV → CUL taxon also relate to cultural identity connected to the environment, sense of place (Masterson et al., 2017), and more generally what has been described as cultural ecosystem services (Millennium Ecosystem Assessment, 2005). For example, Beckage et al. (2018) have modelled the effect of changes in extreme events resulting from climate change on risk perception of individuals. Changes in risk perception may result in changes in emission behaviour given the perceived behaviour of others (social norms) and structural conditions in society, thus feeding back on future climate change.

ENV → CUL also play a role in regional-scale models of poverty traps where decline in natural capital reduces traditional ecological knowledge as a form of cultural capital (Lade et al., 2017b), or in models of human perceptions of local scenic beauty in policy contexts (Bienabe and Hearne, 2006). At the moment, most models deal with these interactions only at a sub-global level. But there is increasing recognition of the need for the more dynamic understanding that formal modelling can provide of such complex psychologically and culturally mediated aspects of human behavior in the Anthropocene (Schill et al., 2019).

## 3.4 MET → MET: economic and socio-metabolic self-interactions

This taxon describes modelled interactions between MET subsystems that connect the material manifestations and artefacts of human societies. Examples include the energy system driving factories, supply chains connecting resource extractors to complex networked production sites or machines constructing infrastructures such as power grids, airports and roads.

Certain processes involving such interactions, e.g. links between the energy system and other sectors such as industrial production, are represented in IAMs in an abstracted, macroeconomic fashion. There exist also agent-based models resolving the dynamics of supply chains that allow to describe the impacts of climate shocks on the global economy in much more detail

(e.g. Otto et al. (2017)). Another class of examples are population models that may include factors such as the influence of income on fertility (Lutz and Skirbekk, 2008). However, to our best knowledge, process-detailed models of the socio-industrial metabolism (Fischer-Kowalski and Hüttler, 1998; Fischer-Kowalski, 2003) or the technosphere (Haff, 2012, 2014) comparable in complexity to biophysical Earth system models have not been published so far.

## 3.5 MET → ENV: greenhouse gas emissions, land-use change and biodiversity loss, impacts on other planetary boundary processes, etc.

This taxon encompasses modelled influences exerted by socio-metabolic subsystems on the biophysical environment including various forms of the "colonisation of nature" (Fischer-Kowalski and Haberl, 1993). Prominent examples in the context of global change and sustainability transformation include human impacts on the environment addressed by the planetary boundaries framework (Rockström et al., 2009a, b; Steffen et al., 2015) such as anthropogenic emissions of greenhouse gases (Stocker et al., 2013), nitrogen and phosphorous, other forms of chemical pollution and novel entities (e.g., nano particles, genetically engineered organisms), land-use change and induced biodiversity loss, exploitation and use of natural resources (Perman, 2003). This taxon also includes various forms of the conversion of energy and entropy fluxes in the biophysical Earth system by human technologies such as harvesting of renewable energy by wind turbines and photovoltaic cells (Kleidon, 2016) or different approaches to geoengineering (Vaughan and Lenton, 2011).

The interactions described by the MET → ENV are central in IAM and ESM studies of the global environmental impacts of human activities in the Anthropocene such as anthropogenic climate change as driven by greenhouse gas emissions and land-use change (Barros et al., 2014; Edenhofer et al., 2014). The latter two key processes are also frequently included in emerging studies of planetary social-ecological dynamics using stylised models (Kellie-Smith and Cox, 2011; Anderies et al., 2013; Heck et al., 2016; Heitzig et al., 2016; Lade et al., 2017a; Nitzbon et al., 2017).

## 3.6 MET → CUL: needs, constraints, etc.

This taxon describes modelled influences and constraints imposed upon socio-cultural dynamics by the material basis of human societies (socio-metabolic subsystems). These include, for example, the effects, needs and constraints induced by the biophysical "hardware" that runs socio-cultural processes: infrastructures, machines, computers, human bodies and brains, and associated availability of energy and other resources. It also includes the effects of technological evolution, revenues generated from economic activity, supply of valued goods, e.g. on opinion formation and behavioural change in the socio-cultural domain, or the consequences of change in demographic distribution of pressure groups on political systems and institutions.

As a recent example, the Beckage et al. (2018) model mentioned above (Sect. 3.3) has one parameter to reflect structural constraints in society that affects the degree to which emission behaviour can be changed. MET → CUL links also appear in models of resource use in social-ecological systems, where social learning of harvesting effort depends on the harvest rate (Wiedermann et al., 2015; Barfuss et al., 2017; Geier et al., 2019) and fish catches influence perceptions about the state of the fishery (Martin and Schlüter, 2015; Lade et al., 2015), or in models of economic impacts on individual voting behaviour (Lewis-Beck and Ratto, 2013).

### 3.7 CUL → CUL: socio-cultural self-interactions

This taxon contains modelled self-interactions between subsystems in the socio-cultural domain that have been described as parts of the noosphere (Vernadsky, 1929/1986), the global subject (Schellnhuber, 1998), or the mental component of the Earth system (Lucht and Pachauri, 2004). Examples include the interaction of processes of opinion dynamics and preference formation on social networks, governance systems and underlying value systems (Gerten et al., 2018) as well as interactions between different institutional layers such as governance systems, formal and informal institutions (Williamson, 1998; Otto et al., 2020a).

Some of these processes related to human behaviour and decision making (Müller-Hansen et al., 2017) have already been studied in models of social-ecological systems on local and regional scales (Schlueter et al., 2012; Schlüter et al., 2017) and have been modelled in various fields ranging from social simulation to the physics of social dynamics (Castellano et al., 2009). However, they are so far largely not included in IAMs of global change or stylised models of planetary social-ecological systems (Verburg et al., 2016; Donges et al., 2017a, b).

### 3.8 CUL → ENV: environmental governance, nature conservation areas, social taboos, sacred places etc.

This taxon encompasses modelled influences that socio-cultural subsystems exert on the biophysical environment. An example for such a class of interactions is environmental governance realized through formal institutions (Ostrom et al., 2007; Folke et al., 2011), where a piece of land is declared as a nature protection area excluding certain forms of land-use which has a direct impact on environmental processes there. Similarly, nature protection areas for biodiversity conservation have been represented in marine reserve models (Gaines et al., 2010). Another related example for CUL → ENV links are nature-related values and informal institutions such as respecting sacred places in the landscape and following social taboos regarding resource use (Colding and Folke, 2001). Different forms of environmental governance have been modelled via so-called decision or sustainability paradigms (Schellnhuber, 1998; Barfuss et al., 2018; Heitzig et al., 2018).

Direct CUL → ENV links arguably cannot be found in the real world, in that socio-cultural influences on environmental processes must be mediated by their physical manifestations in the socio-metabolic domain (e.g. in the case of nature protection areas through the constrained actions of resource users, government enforcement efforts and infrastructures such as fences). However, such direct CUL → ENV links may be implemented in models, even on the global scale, such as in trade-off assessments of multiple land-uses (e.g. Boysen et al. (2017); Phalan (2018)).

### 3.9 CUL → MET: socio-economic policies and governance choices, value-driven consumption, etc.

Finally, this taxon contains modelled links pointing from socio-cultural to socio-metabolic subsystems. Examples include socio-economic policies and governance choices such as taxes, regulations or caps that influence the economy (e.g. carbon caps or taxes in the climate change mitigation context) or demographics (e.g. family planning and immigration policies) as well as the physical manifestations of financial market dynamics such as real estate bubbles. CUL → MET interactions

also encompass the influence of cultural values, norms and lifestyles on economic demand and consumption and consequent changes in industrial production, building, transportation and other sectors.

Policy measures such as taxes, regulations or caps are much studied by IAMs of anthropogenic climate change (Edenhofer et al., 2014), while influences of value and norm change on economic activities such as general resource use (Wiedermann et al., 2015; Barfuss et al., 2017; Geier et al., 2019) and fishing (Martin and Schlüter, 2015; Lade et al., 2015) has been studied in the social-ecological modelling literature, but at a mostly local to regional level.

### 3.10 Higher-order taxonomies of feedback loops and more complex interaction networks

Beyond the taxonomy of interactions introduced above, higher-order taxonomies could also be derived. For example, a taxonomy of feedback loops can be derived from the $3 \times 3$ taxonomy of links, leading to six taxa for feedback loops of length two in models of World-Earth systems: given a pair of interaction taxa $A \to B$ and $B \to A$, the resulting taxon for loops between $A$ and $B$ may be denoted as $A \circlearrowleft B$. Many such feedback loops relevant for sustainability are not or only rigidly treated in current ESMs and IAMs. For example, the ENV $\circlearrowleft$ MET feedback loop is typically not sufficiently represented in IPCC-style analyses, because the impacts of climate change on human societies are not explicitly modelled or ill-constrained in IAMs (Sect. 3.5). Furthermore, feedback loops of the type CUL $\circlearrowleft$ X, where X may be subsystems from ENV, MET or CUL are mostly missing altogether, not the least because CUL is not represented, or only fragmentarily included, in current ESMs and IAMs.

Longer and more complex paths and subgraphs of causal interactions between subsystems could be classified by further higher-order taxonomies (e.g. inspired by the study of motifs, small subgraphs, in complex network theory, Milo et al. (2002)). This approach quickly leads to a combinatorial explosion, e.g. for 3-loops of the type $A \to B \to C \to A$ involving three modelled subsystems $A, B, C$ and their interactions enumeration and counting of all possible combinations shows that there are already 11 distinct taxa for feedback loops of this kind. However, there are systematic methods available for classifying and clustering causal loop diagrams that could be leveraged to bring order into more complex models of World-Earth systems (Van Dijk and Breedveld, 1991; Rocha et al., 2015). Overall, such higher-order taxonomies could help in the design of models or model suites that can deal with different aspects of (nonlinear) interactions between World-Earth subsystems and serve as tools for understanding the emergent co-evolutionary macrodynamics.

## 4 An exemplary model showing complex co-evolutionary dynamics in a World-Earth system

At present, to our best knowledge, process-detailed World-Earth models that are comprehensive in the sense of the proposed taxonomies are not available. Therefore, in this section, we give an illustrative example of a stylised World-Earth system model that covers all classes of real-world processes that appear relevant in major global feedbacks. Even such a very simple World-Earth system model can contain a social-ecological feedback loop involving subsystem interactions introduced above (Sect. 3), and leading to a biophysical Earth system dynamics that depends crucially on a social-cultural evolution and vice versa. We also demonstrate how the taxonomies described above can be applied to classify model components and reveal the interaction

structures that are implicit in the model equations. The companion paper of this article applies the taxonomies to develop a more complex illustrative World-Earth model using the copan:CORE framework (Donges et al., 2020).

The example model studied here, copan:DISCOUNT, describes a world where climate change drives a change of countries' value systems, represented here just by the long-term discount factors their governments use in policy-making, which can be 5 interpreted as their relative interest in future welfare as opposed to current welfare. These discount factors drive countries' emissions and thus in turn drive climate change, represented by a global atmospheric carbon stock. While the detailed description of the model's assumptions below will make clear that this causal loop involves eight of the nine interaction taxa shown in Fig. 2, the model is so designed that the description of the resulting dynamics from all these interactions can be reduced to just two ordinary differential equations, one for the fraction of "patient" countries and one for atmospheric carbon stock. The 10 novelty of this model is that it endogenises socially transmitted choice of discount rates in a greenhouse gas emissions game to illustrate the effects of social-ecological feedback loops that are so far typically not considered in current climate economics and IAM modelling efforts.

The aim of this particular model design is to show clearly that while the taxonomy developed in this paper aims at being helpful in designing and analysing World-Earth models, this does not mean the different taxa need always be easily identifiable 15 from the final model equations.

Before relating its ingredients to the introduced taxa, let us describe the model without referring to that classification. In our model, we assume that each country's metabolic activities are guided by a trade-off between the undesired future impacts of climate change caused by global carbon emissions, and the present costs of avoiding these emissions domestically. Similar to the literature on international environmental agreements and integrated assessment modelling, this tradeoff is modelled as a 20 non-cooperative game between countries applying cost-benefit optimisation. The tradeoff and hence the evolution of the carbon stock is strongly influenced by the discount factor $\delta$ that measures the relative importance a country assigns to future welfare as compared to present welfare. The higher $\delta$, the more a country cares about the future and the more they will reduce their emissions in order to avoid future climate impacts. While the economic literature treats $\delta$ as an exogenous parameter that has to be chosen by society (e.g., Arrow et al. (2013)), our model treats $\delta$ as a social trait that changes in individual countries over 25 time because countries observe each other's welfare and value of $\delta$ and may learn what a useful $\delta$ is by imitating successful countries and adopting their value of $\delta$. Because of the existence of climatic tipping points, this social dynamics does not only influence the state of the climate system but is in turn strongly influenced by it. Depending on whether the system is far from or close to tipping points, the trade-off between emissions reduction costs and additional climate damages can turn out quite differently and different values of $\delta$ will be successful.

30 Let us now present and decompose the model's basic causal loop in terms of the above introduced taxonomy, as shown in Fig. 3, starting in the central box. The countries' metabolisms (MET) combust carbon (MET → MET), leading to emissions (MET → ENV) that increase the global atmospheric carbon stock $C$ (ENV), part of which is then taken up by other carbon reservoirs (ENV → ENV). $C$ increases global mean temperature, leading to climate change (ENV → ENV) and thus to future climate impacts (i) on the countries' metabolisms (ENV → MET) and (ii) on aspects of the environment people care about, 35 such as biodiversity (ENV → ENV → CUL). Countries evaluate these expected damages (MET → CUL; ENV → CUL)

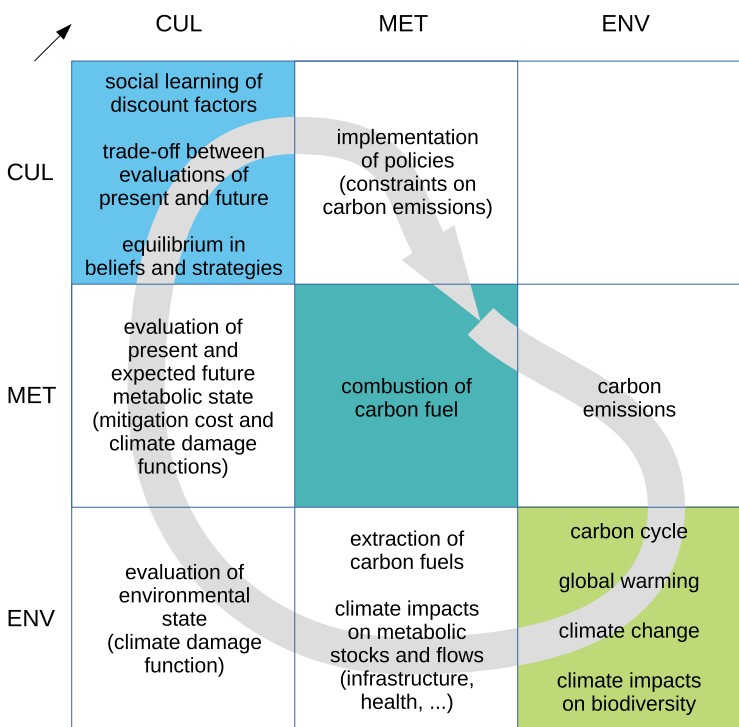

|  | CUL | MET | ENV |
|---|---|---|---|
| **CUL** | social learning of discount factors<br><br>trade-off between evaluations of present and future<br><br>equilibrium in beliefs and strategies | implementation of policies (constraints on carbon emissions) | |
| **MET** | evaluation of present and expected future metabolic state (mitigation cost and climate damage functions) | combustion of carbon fuel | carbon emissions |
| **ENV** | evaluation of environmental state (climate damage function) | extraction of carbon fuels<br><br>climate impacts on metabolic stocks and flows (infrastructure, health, ...) | carbon cycle<br><br>global warming<br><br>climate change<br><br>climate impacts on biodiversity |

**Figure 3.** Planetary social-ecological processes and interactions represented in the copan:DISCOUNT model displayed in matrix form following Fig. 2. The co-evolutionary cycle of dynamic interdependencies implemented in the model is indicated by the grey arrow.

and the costs of avoiding emissions (MET → CUL), use their respective discount factors (CUL), which they learn by imitation (CUL → CUL), to assess possible domestic emissions constraints, then reach a strategic equilibrium with other countries (CUL → CUL) and implement the chosen emissions constraints (CUL → MET), this closing the long loop.

In the statistical limit of this model for a large number of countries, derived in detail in the Appendix A, this complex feedback dynamics is nicely reduced to just two equations,

$$\dot{C} = E_0 - c\, s(C)\, \phi(F) - rC, \tag{1}$$

$$\dot{F} = \ell F(1 - F)[P(D(C,F)) - P(-D(C,F))], \tag{2}$$

where $C$ is excess atmospheric carbon stock and $F$ the fraction of "patient" countries (those that apply a large value of $\delta$), and where $s(C)$ is a damage factor, $\phi(F)$ is a certain linear transformation of $F$, $D(C,F)$ is the utility difference between a country using discount factor $\alpha$ and a country using $\beta$, and $P(D)$ is a resulting imitation probability, all these derived in detail in the Appendix A. Some of the various terms in these formulas can be classified clearly as belonging to one taxon, e.g., BAU emissions $E_0$ belong to MET → ENV, carbon-uptake $-rC$ to ENV → ENV, and the imitation probability $P(D)$ to CUL →

CUL. But others cannot, e.g., certain terms occurring in the formula for $D$ combine climate damages $s(C)$ (ENV $\rightarrow$ MET $\rightarrow$ CUL) with countries' values systems, represented by $\phi(F)$ (CUL). The dynamics are governed by about a dozen parameters controlling the relative speeds and intensities of subprocesses, costs and benefits of emissions reductions, and details of the learning-by-imitation process, as described in the Appendix (Sect. A).

Let us analyse a typical dynamics of the model, shown in Fig. 4, and relate it again to our taxonomy of subsystem interactions. Consider the middle green trajectories in the lower panel starting at a low atmospheric carbon stock of $C = 1$ (fictitious units) and a medium fraction of patient countries of $F = 0.5$ (green dot). At this point, both patient and impatient countries evaluate the state of the world very similarly, hence not much imitation of discount factors happens (weak CUL $\rightarrow$ CUL dynamics), so that $F$ may fluctuate somewhat but is not expected to change much. At the same time, as the climate damage curve (middle

panel) is still relatively flat, global emissions are higher than the natural uptake rate (strong MET $\rightarrow$ ENV influence), and $C$ is likely to increase to about 1.7 without $F$ changing much. During this initial pollution phase, climate damages increase (the ENV $\rightarrow$ MET/CUL links becomes stronger) and the slope of the damage curve increases as more climatic tipping points are neared or crossed. This decreases the patient countries' evaluations faster than the impatient countries', hence patience becomes less attractive and countries fatalistically decrease their discount factor, so that $F$ declines to almost or even exactly zero (the

CUL $\rightarrow$ CUL dynamics becoming first stronger then weaker again) while $C$ grows to about 3.0. In that region, most tipping points are crossed and the damage curve flattens again, causing the opposite effect, i.e., making patience more attractive. If the idea of patience has not "died-out" at that point (i.e., $F$ is still $> 0$), discount factors now swing to the other extreme with $F$ approaching unity (CUL $\rightarrow$ CUL dynamics becoming temporarily very strong), shown by one green trajectory, while emissions are first almost in equilibrium with natural carbon uptake at about $C = 3.2$ (weak MET $\rightarrow$ ENV effect) and then decline ever

faster once the vast majority of countries got patient (stronger MET $\rightarrow$ ENV). This trajectory finally converges to the stable steady state at a low carbon stock of about $C = 1.5$ and $F = 1$. Note that there is also some small probability that this point is reached much faster without the long detour if the stochastic social dynamics at the starting point give patience a random advantage, as on two of the plotted trajectories.

     As is typical in models with various interactions, changes in their relative interaction rates can cause highly nonlinear and

even qualitative changes in model behaviour. A comparison of the top and bottom panels in Fig. 4 (see also its caption) shows that this is in particular true for World-Earth models when the rates of socio-cultural processes of the CUL $\rightarrow$ CUL type are changed (as can be claimed is indeed happening in reality since the middle of the 20th century). It should be emphasised again that these socio-cultural processes are specifically those that are least or not at all represented in current models of global change, pointing to the necessity and expected progress in understanding when including them in more comprehensive

World-Earth models.

     Overall, the DISCOUNT model provides a first test of the taxonomy's guiding principles. It demonstrates the taxonomy's operative capacity to trace links between established dynamical systems methodology and macro behaviour; it is compatible with diverse research fields, here linking, among others, carbon cycles and social learning; and it has appropriate compactness, since tracing the loops and flows between taxa in this World-Earth model do not make us need to rethink the whole structure

of the taxonomy.

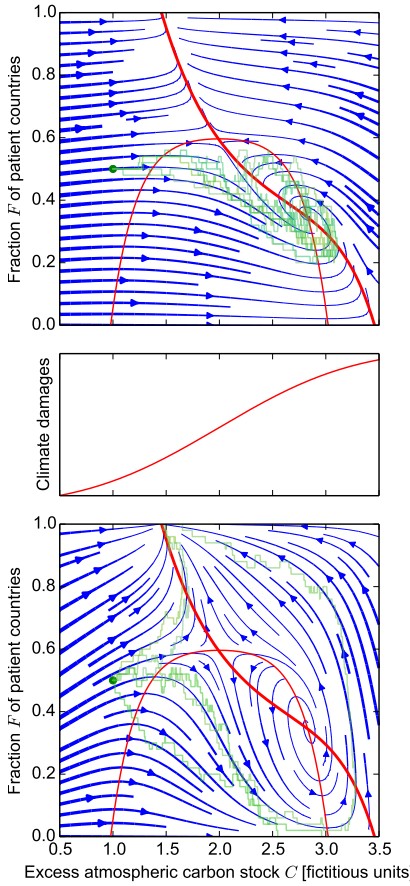

**Figure 4.** Typical dynamics of the copan:DISCOUNT model of the co-evolution of the global atmospheric carbon stock $C$ and the time preferences of countries, represented by the fraction $F$ of patient countries. Of five simulated stochastic trajectories (top and bottom panel, green lines) starting at the same initial state (green dot), some will converge fast to the more desirable stable steady state at $C \approx 1.5$, $F = 1$ where climate damages (middle panel) are still relatively low, while other trajectories will approach the less desirable focus point (spiralling steady state) at $C \approx 2.8$, $F = 0.35$ where climate damages are relatively high. Depending on whether countries adjust their time preferences slowly (top panel) or fast (bottom), that focus point is either a stable attractor catching most trajectories that come near it (top) or an unstable repeller which many trajectories have to compass to approach the desirable state after a long transient detour of high damages (bottom). Blue lines show the average development represented by two ordinary differential equations (see Appendix A for details), red lines are the corresponding nullclines (thin: $\dot{F} = 0$, thick: $\dot{C} = 0$), and their other intersection at $C \approx 2$, $F \approx 0.6$ is a saddle point. Parameters: $E_0 = 1.6$, $c = 1$, $r = 0.45$, $l = 0.2$ (top) or $1.3$ (bottom), $\gamma = 1.1$, $\mu = 2$, $\sigma = 1$, $\beta = 0.1$, $\alpha = 0.5$, $G = 2$, $N = 50$, $p_0 = 0.5$, $q = 3$.

# 5 Conclusions

In this article, we have presented a taxonomy of processes and co-evolutionary interactions in models of World-Earth systems (i.e. planetary-scale social-ecological systems). For reasons of compactness and compatibility with existing research fields and methodologies we have proposed three taxa for modelled subsystems, and furthermore described a classification of modelled interactions between subsystems into nine taxa. We have illustrated the clarity that this taxonomic framework confers, using a stylised model of social-ecological co-evolutionary dynamics on a planetary scale that includes explicitly socio-cultural processes and feedbacks.

We argue that a relatively simple taxonomy is important for stimulating the discourse on conceptualisations of World-Earth systems. It can help with operational model development as is illustrated by the work reported in the companion paper (Donges et al., 2020). The proposed taxonomy can also help in interdisciplinary communication, model critique, and potentially even participatory modelling processes by providing an organisational scheme and a shared vocabulary to refer to the different components that need to be brought together. However, we acknowledge that alternative, more detailed taxonomies can be beneficial in more specialised settings, e.g. ecological processes are now subsumed in the biophysical taxon, but it may be useful to distinguish them from the geophysical for a clearer understanding of interactions with the socio-metabolic taxon. In other contexts, it may be useful to establish a socio-epistemic taxon separate from the socio-cultural taxon for describing subsystems, processes and interactions involving, for example, symbolic representations and transformations of knowledge through science and technology (Renn, 2018). Along these lines, our framework may be helpful as a blueprint for constructing such alternative, possibly more detailed taxonomies.

Throughout the paper, we have illustrated the taxonomic framework using examples of subsystems, processes and interactions that are already represented in mathematical and computer simulation models in various disciplines. We have not attempted to provide a comprehensive classification of all such modelling components that would be relevant for capturing future trajectories of World-Earth systems in the Anthropocene. Neither have we addressed dynamics beyond the reach of current modelling capabilities, such as long-term evolutionary processes acting within the biophysical taxon or broad patterns and singularities in the dynamics of technology, science, art and history (Turchin, 2008). But we have shown the merits of epistemological pluralism, to enable productive dialogue and interaction between the diversity of World modelling approaches and the biophysical Earth representations that exist and that have agency in a Latourian sense, e.g. through the IPCC processes.

Applying the proposed taxonomy reveals relevant directions in the future development of models of global change to appropriately represent the dynamics of up to planetary-scale social-ecological systems in the Anthropocene. Regarding the sticky problem of representing causality in such a complex system, every possible contributory model is a Pandora's box out of which theoretical controversies and cross-disciplinary battles emerge. The taxonomy outlined here at least partly illuminates what is in this box, making it easier to have more open discussions among modellers about their theories and hypotheses about causality.

While current Earth System Models focus exclusively on representing biophysical subsystems and their interactions and Integrated Assessment Models capitalise on those in the socio-metabolic taxon, socio-cultural subsystems and processes such

as the dynamics of opinions and social networks, behaviours, values and institutions and their feedbacks to biophysical and socio-metabolic subsystems remain largely uncovered in planetary-scale models of global change. Integrating these decisive dynamics in World-Earth Models is a challenging, but highly promising research programme (Schellnhuber, 1998, 1999; Steffen et al., 2020) comparable to the development of biophysical Earth system science in the past decades following the foundational blueprints of Bretherton et al. (1986, 1988). We use the copan:DISCOUNT model to demonstrate the value of the taxonomy for tracing how dynamics and feedbacks loop through different taxa, enabling better model design and communication about path-breaking approaches to World-Earth modelling. Following this track will help to develop models that go beyond a climate-driven view of global change and to bridge the "divide" that keeps being spotlighted as the problematic hyphen in prevalent social-ecological/human-nature/etc system concepts. It will also contribute to a deeper understanding of the functioning of complex World-Earth systems machinery in the Anthropocene. By supporting the development and discussion of new family of models, and not by pushing for a rigid and universalising model of everything, applying the taxonomy promises to yield important insights on well-designed policy interventions to foster global sustainability transformation, build World-Earth resilience and avoid social-ecological collapse.

**Code availability**

A Python script for integrating and analysing the copan:DISCOUNT model is available at github.org/pik-copan/pycopandiscount(DOI: 10.5281/zenodo.4704936).

**Author contributions**

JFD designed and coordinated the research. JFD led the writing of the manuscript with strong contributions from WL, SC, and JH. JH developed the copan:DISCOUNT model, performed model simulations and analysed results. All other authors contributed to the writing of the paper and the development and discussion of the presented taxonomic framework.

**Competing interests**

The authors declare that they have no conflict of interest.

**Special issue statement**

This article is part of the special issue "Social dynamics and planetary boundaries in Earth system modelling". It is not associated with a conference.

*Acknowledgements.* This work has been carried out within the framework of PIK's project on Coevolutionary Pathways in the Earth system (COPAN). It was supported by the Stordalen Foundation via the Planetary Boundary Research Network (PB.net), the Earth League's Earth-

Doc programme, the Leibniz Association (project DOMINOES), the Federal Ministry for Education and Research (BMBF, project GLUES), Humboldt University, Berlin, the Swedish Research Council Formas (Project Grant 2014-589), a core grant to the Stockholm Resilience Centre by Mistra, the Heinrich Böll Foundation, the European Research Council (ERC) under the European Union's Seventh Framework Programme (FP/2007-2013)/ERC grant agreement no. 283950 SES-LINK) and under the European Union's Horizon 2020 Research and Innovation Programme (ERC grant agreement No. 682472 – MUSES and ERC grant agreement No. 743080 ERA). We are thankful to the participants of the LOOPS 2014 workshop on "Closing the loop – Towards co-evolutionary modelling of global society-environment inter-actions", held at Kloster Chorin, Germany and the LOOPS 2015 workshop on "From limits to growth to planetary boundaries: defining the safe and just space for humanity" held in New Forest, Southampton, UK for inspiring discussions that sparked this paper. Ilona M. Otto, Marc Wiedermann and Finn Müller-Hansen are acknowledged for helpful comments on ideas presented in this paper. We are grateful for the excellent quality of creative stimulus prompted by the comments of Carsten Herrmann-Pillath, Birgit Müller and an anonymous reviewer.

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

## Appendix A: The copan:DISCOUNT model

The illustrative model copan:DISCOUNT simulates the co-evolution of $C \geqslant 0$, the excess global atmospheric carbon stock above an equilibrium value that would be attained for zero GHG emissions, and the fraction $F \in [0, 1]$ of the world's countries that care strongly about their future welfare. While $C$ represents the macroscopic state of nature, $F$ represents the macroscopic state of the global human society.

As the derivation of the model below will show, the time evolution of $C$ and $F$ is eventually given by Eqs. 1 and 2. Their
governing parameters are business-as-usual emissions $E_0 > 0$, an abatement cost factor $c > 0$, a carbon uptake rate $r > 0$, a learning rate $\ell > 0$, a damage coefficient $\gamma > 0$, a mean tipping point location $\mu > 0$ and spread $\sigma > 0$, two candidate discount rates $0 < \beta < \alpha < 1$, an economic growth factor $G \geqslant 1$, the total number of countries $N > 0$, a curiosity parameter $0 < p_0 < 1$, and a myopic rationality parameter $q > 0$. The equations are derived by combining a standard emissions game model from the literature on international environmental agreements (Barrett, 1994) with a social imitation dynamics that governs the evolution
of the countries' time discounting factors as follows.

### A1    Countries, welfare

At each point in continuous time, $t$, a number of $N > 1$ similar countries, $i$, choose their individual *abatement levels* (carbon equivalents per time), $a_i(t) \geqslant 0$. Global abatement and carbon emissions per time (an interaction of type MET $\rightarrow$ ENV) are

then

$$A(t) = \sum_{i=1}^{N} a_i(t), \qquad\qquad E(t) = E_0 - A(t), \qquad\qquad (A1)$$

where $E_0 > 0$ are global "business-as-usual" emissions.

Country $i$ chooses $a_i(t)$ rationally but myopically, only taking into account its own welfare in the present and in "the future" (after a fixed time interval of, say, fifty years). Its present welfare, $W_i^0(t)$, is given by some business as usual welfare, normalised to unity, minus the costs of emissions reductions (MET $\rightarrow$ CUL), which are a quadratic function of $a_i(t)$ as usual in stylised models of international environmental agreements (Barrett, 1994),

$$W_i^0(t) = 1 - \frac{a_i(t)^2}{2c/N}, \qquad\qquad (A2)$$

where $c/N > 0$ is a cost parameter that is normalised with $N$ to make the Nash equilibrium outcome (see below) independent of $N$.

Country $i$'s "future" welfare (belonging to MET), $W_i^1(t)$, is a higher business-as-usual welfare given by a growth parameter $G > 1$, minus the value of additional damages from climate change caused by the present emissions, which are a linear function of $E(t)$:

$$W_i^1(t) = G - s(C(t))E(t), \qquad\qquad (A3)$$

where $s(C(t)) > 0$ is a *damage factor* that depends on the current carbon stock (see below). Note that while these additional damages $s(C)E(t)$ caused by the present emissions, total damages will still be a nonlinear function of stock $C$ since the factor $s(C)$ changes with $C$, representing the presence of tipping points (see below).

## A2  Discounting, emissions

Since $W_i^1$ increases in $a_i$ while $W_i^0$ decreases, choosing an optimal value for $a_i$ involves a trade-off between present and future welfare, which we assume is done in the usual way by using some current *discount factor* $0 < \delta_i(t) < 1$ (an element of taxon CUL) that measures the relative weight of future welfare in country $i$'s optimisation target ("utility") at time $t$, $U_i(t)$:

$$U_i(t) = (1 - \delta_i(t))W_i^0(t) + \delta_i(t)W_i^1(t). \qquad\qquad (A4)$$

For simplicity, we assume that only two different discount factors are possible, $0 < \beta < \alpha < 1$, and call a country with $\delta_i(t) = \alpha$
"patient", so that the state of global society at time $t$ can be summarised by the fraction $F(t)$ of patient countries:

$$F(t) = |\{i : \delta_i(t) = \alpha\}|/N. \qquad\qquad (A5)$$

Given carbon stock $C(t)$ (ENV) and discount factors $\delta_i(t)$, the countries thus face a simultaneous multi-agent multi-objective optimisation problem, each $i$ trying to optimise their utility

$$U_i(t) = \left(1 - \delta_i(t)\right)\left(1 - \frac{a_i(t)^2}{2c/N}\right)$$

$$+ \delta_i(t)\left(G - s\big(C(t)\big)\right)\left(E_0 - \sum_{j=1}^{N} a_j(t)\right). \tag{A6}$$

by choosing $a_i(t)$. As in the literature on international environmental agreements, e.g., Barrett (1994), we assume this is solved by making the choices independently and non-cooperatively, i.e., putting $\partial U_i(t)/\partial a_i(t) = 0$ for all $i$ simultaneously, leading to a system of $N$ equations whose solutions $a_i(t)$ form the Nash equilibrium choices (CUL $\to$ CUL),

$$a_i(t) = \frac{c}{N}\frac{\delta_i(t)}{1 - \delta_i(t)}s(C(t)), \tag{A7}$$

$$U_i(t) = 1 + \delta_i(t)(G - E_0\,s(C(t)) + c\,s(C(t))^2\phi(F(t)) - 1)$$

$$- \frac{c}{2N}\frac{\delta_i(t)^2}{1 - \delta_i(t)}s(C(t))^2 \tag{A8}$$

and the aggregate abatement (CUL $\to$ MET) and emissions

$$A(t) = s(C(t))\,c\,\phi(F(t)), \qquad\qquad E(t) = E_0 - A(t), \tag{A9}$$

where

$$\phi(F(t)) = F(t)\frac{\alpha}{1 - \alpha} + (1 - F(t))\frac{\beta}{1 - \beta}. \tag{A10}$$

## A3   Evolution of discount factors

While economic models treat the discount factor of a country as an exogenous parameter, we assume that the value of $\delta_i$ is a social trait that may be changed over time due to the observation of other countries' discount factors and their resulting utility (CUL $\to$ CUL). As in many models of the spread of social traits (e.g., Traulsen et al. (2010); Wiedermann et al. (2015)), we assume that each country $i$ may adopt another country $j$'s value of $\delta$ (social learning by imitation) and that the probability $P$ for doing so depends on the difference between $i$ and $j$'s current utility, $D_{ij}(t) = U_j(t) - U_i(t)$, in a nonlinear, sigmoid-shaped fashion, with $P(D) \to 0$ for $D \to -\infty$ and $P(D) \to 1$ for $D \to \infty$. The utility difference between a country using $\alpha$ and a country using $\beta$ is

$$D(t) = [\alpha - \beta](G - E_0 s(C(t)) + cs(C(t))^2\phi(F(t)) - 1)$$

$$- \left[\frac{\alpha^2}{1 - \alpha} - \frac{\beta^2}{1 - \beta}\right]\frac{cs(C(t))^2}{2N}. \tag{A11}$$

This difference is zero iff the discounting summary statistics $\phi(F(t))$ equals

$$\phi_F(C(t)) := \frac{\frac{\alpha^2}{1-\alpha} - \frac{\beta^2}{1-\beta}}{2N[\alpha - \beta]} + \frac{E_0}{cs(C(t))} - \frac{G - 1}{cs(C(t))^2} \tag{A12}$$

Since $\alpha > \beta$, we have $D(t) > 0$ iff $\phi(F(t)) < \phi_F(C(t))$, meaning that depending on the stock and the fraction of patient countries, either patience or impatience might be more attractive, so that one can expect interesting learning dynamics.

We assume that at each point in time, each country $i$ independently has a probability rate $\ell > 0$ to perform a "learning step". If $i$ does perform a learning step at time $t$, it compares its current utility $U_i(t)$ with that of a randomly drawn country $j$ and sets its discount factor $\delta_i(t)$ to the value of $\delta_j(t)$ with a probability given by the generalised logistic function,

$$P(D_{ij}(t)) = \frac{1}{1 + \frac{1-p_0}{p_0} \exp(-\frac{q}{p_0(1-p_0)} D_{ij}(t))}, \tag{A13}$$

where $0 < p_0 < 1$ and $q > 0$ are parameters so that $P(0) = p_0$ and $P'(0) = q$.

The "curiosity" parameter $p_0$ can be interpreted as a measure of a country's curiosity-driven exploration of a different discount factor without expecting a welfare increase. The larger $p_0$, the more frequent switches will occur, but in both directions between the two candidate discount rates, mainly generating more variance and fluctuations that can be seen as a form of "noise". The "myopic rationality" parameter $q$ can be interpreted as a measure of a country's rationality, because the probability of switching to the other country's discount rate is higher if the other country has higher welfare (and zero if that is not the case) – but it is a myopic rationality, because the agent only takes its present welfare into account. The larger $q$, the faster discount factors will converge to the one currently generating the largest welfare.

To get a deterministic evolution that can be represented by an ordinary differential equation, we only track the *expected* fraction $F(t)$ of patient countries, which evolves as

$$\dot{F}(t) = \ell F(t)(1 - F(t))[P(D(t)) - P(-D(t))], \tag{A14}$$

while the actual number of patient countries would follow a stochastic dynamics involving binomial distributions that converges to the above in the statistical limit $N \to \infty$. Note that $\dot{F}(t) = 0$ iff $F(t) \in \{0,1\}$ or $\phi(F(t)) = \phi_F(C(t))$.

## A4  Carbon stock, damage factor

For ease of presentation, we drop the denotation of time dependence from here on. We assume that the atmospheric carbon stock evolves according to a simplistic dynamics involving only emissions and carbon uptake by other carbon stocks,

$$\dot{C} = E - rC = E_0 - cs(C)\phi(F) - rC \tag{A15}$$

with a constant *carbon uptake rate* $r > 0$ (ENV $\to$ ENV). Note that $\dot{C} = 0$ iff $\phi(F)$ equals

$$\phi_C(C) = \frac{E_0 - rC}{cs(C)}. \tag{A16}$$

In order that $C \geqslant 0$ for all times, we require that $\dot{C} \geqslant 0$ whenever $C = 0$, which is ensured by assuming that the parameters fulfil $E_0 \geqslant c\gamma \exp(-\mu^2/2\sigma^2)\phi_1$ where $\phi_1 = \alpha/(1-\alpha)$.

We further assume that $s(C)$, the value (MET $\to$ CUL; ENV $\to$ CUL) of the additional damages from climate change (ENV $\to$ MET; ENV $\to$ CUL) due to a marginal increase in emissions at an existing carbon stock $C$ (ENV $\to$ ENV), is a positive

function of $C$ that has a unique maximum at some critical stock $\mu$ at which small changes in stock lead to large changes in damages due to the presence of tipping points. To approximate a damage function that is a sum of a number of sigmoid-shaped functions representing individual tipping points whose locations and amplitudes are roughly normally distributed, we take $s(C)$ to be Gaussian,

$$s(C) = \gamma \exp(-(C-\mu)^2/2\sigma^2), \tag{A17}$$

with parameters $\gamma > 0$, $\mu > 0$, $\sigma > 0$. This completes our derivation of the two ordinary differential equations for $C$ and $F$.

## A5 Steady states, stability

We can distinguish three types of steady states where $\dot{C} = \dot{F} = 0$.

(1) All countries are impatient, $F = 0$ (which implies $\phi(F) = \phi_0 := \beta/(1-\beta)$), and $(E_0 - rC)/cs(C) = \phi_0$. The latter is equivalent to $c\phi_0\gamma \exp(-(C-\mu)^2/2\sigma^2) = E_0 - rC$ which has generically one or three solutions in $C$ with $C > 0$. If there are three, the middle one is always unstable. The others are stable iff $D < 0$.

(2) All countries are patient, $F = 1$ (which implies $\phi(F) = \phi_1$) and $(E_0 - rC)/cs(C) = \phi_1$. The latter is equivalent to $c\phi_1\gamma \exp(-(C-\mu)^2/2\sigma^2) = E_0 - rC$ which again has generically one or three solutions in $C$ with $C > 0$. Again, if there are three, the middle one is always unstable. Again, the others are stable iff $D < 0$. The possibility of two stable states with $F = 1$, one with a small and one with a large $C$, indicates that even if all countries eventually become patient, this may happen too slowly to prevent a level of climate change (large $A$) that makes ambitious mitigation even for patient countries too costly in view of the small amount of climate damages that could then still be avoided.

(3) $0 < F < 1$ and $\phi(F) = \phi_F(C) = \phi_C(C)$. This has at most four different solutions in $C$ with $C > 0$, to each of which corresponds at most one solution in $F$. We know of no simple conditions for assessing their stability but from our numerical experiments we conjecture that (i) at most one of them is stable, namely the one with the largest $C$, (ii) its stability depends only on the learning rate $\ell$, being stable up to a critical value $\ell^*$, then unstable; (iii) For $\ell < \ell^*$, it is a stable focus and the leftmost steady state with $F = 0$ is unstable. Hence at most four stable steady states can exist: at most two with $F = 1$, and either at most two with $F = 0$ or at most one with $F = 0$ plus the stable focus with $0 < F < 1$.