# Peer review of "Taxonomies for structuring models for World-Earth systems analysis of the Anthropocene: subsystems, their interactions and social-ecological feedback loops"

_Earth System Dynamics, 2018_

## Referee Comment (RC1) · C. Herrmann-Pillath (Referee) · 5 May 2018

This is a timely and much-needed effort at creating more realistic and inclusive/integrative models of the Earth system that meet the challenge to model the complexities of human action in various systemic contexts (such as economy, society and so on). It is far too easy to criticize such endeavours because they necessarily offer many points of attack, being the first steps in this direction. This would be unfair. In my review, I try to find a proper balance between necessary criticism on the one hand, and endorsement and encouragement on the other hand. I am writing from the position

of an economist working in the field of evolutionary, institutional and ecological eco-
nomics (hence, clearly 'heterodox') who always injects a strong dose of philosophy in
his work. That is, I am an outsider to the modelling community in the Earth and Climate
Sciences. I will always combine general observations with specific comments on the
discount model. Yet, my focus is on principled issues, hoping that this might provide
different perspectives than those within the modelling community. The authors explic-
itly recognize the philosophical dimensions of their work; hence I start out from there,
and I congratulate them for showing the grit to use the notion of 'World-Earth system'
(p. 2). However, this immediately raises many philosophical questions. 1. In which
sense can we separate 'Earth' from 'World'? Although we must assume that there is
an 'Earth' interacting with 'World', the 'Earth' is only accessible via the sciences which
construct it as a 'World'. That means, we must recognize the fundamental fact of our
incomplete knowledge about 'Earth'. In other words, a complete model of the 'World-
Earth system' must be reflexive in the sense of including the Earth Sciences (and
others) as endogenous generator of our scientific conceptions of 'Earth'. There are
direct implications for the subsystems: For example, our knowledge about the biophys-
ical mechanisms will impact on public perceptions of policy issues, and so on (which
the authors aim at modelling explicitly, indeed). These will feedback on funding Earth
Systems sciences, and hence will determine how 'Earth' will appear to us in future
times. I think that science is an essential part of the 'social-cultural taxon' and cannot
just be imagined as being exogenous. But if it is endogenous, 'Earth' is endogenous.
Looking at the discount model, this seems implicit to the role of the parameters p and
q. Although these are essential for driving the specific model dynamic in determining
the probability of switching discount rates, they are not explained in any way. To be
frank, I have difficulties in understanding their meaning. What exactly is 'curiosity' of
a country? What is 'myopic rationality'? Many people would just think that the latter
means, well, steep discounting of the future. Another definition would be the length of
the time horizon, which differs from discount rates applied within that horizon. Thus,
besides from the need to give precise definitions these parameters and explain how

they could be measured, I suggest that they refer to the endogeneity of science. Just consider Trumpian America: the recent issue of 'The Economist' has an article about 'Swamp Science' at the EPA. This is actively reducing 'curiosity', it seems to me. In other words, I think that if one uses the 'World-Earth' duality, one faces the challenge of treating human knowledge about 'Earth' as endogenous. There is no external standpoint of the model-builder. Another excellent example for this problem is the treatment of 'damage' and 'welfare' in the discount model. It seems that the authors think that there is an objective measure of welfare and damage. But we know that this is one of the most difficult and disturbing aspects of IAM, namely that the damage function is endogenous and depends on the discount rate. That is why some economists now even take the very radical step to build their models without damage function (Llavador, H., J. E. Roemer und J. Silvestre (2015): Sustainability for a Warming Planet. Cambridge University Press). But behind this is the simple, but deeply philosophical problem that neither welfare nor damage can be assessed from the standpoint of an external observer. Apparently, the authors are aware of this, as on p. 17 we find the expression: s(C) (ENV → MET → CUL). But that implies that social learning may not only happen via imitation of discount rates, but also via diffusion of valuations, or, 'worldviews'. 2. The next foundational question is whether there is 'ONE World', which seems implicated by putting both World and Earth together into one 'system'. Obviously, this is not just annoying sophism: There are many philosophers who cast doubt on that (just mention the German philosopher Markus Gabriel with the provocative book title 'Why there is no World' 'Warum es die Welt nicht gibt'). The serious argument behind it ties up with the previous: Science constructs worlds, and there is no necessity that these are just 'one'. This is even more evident if we consider human worlds: Actually, it is the CUL taxon that creates the 'worlds'. I think that it would be most helpful for the authors to look at Bruno Latour's recent work on Gaia and the more general work on 'modes of existence' (see http://modesofexistence.org/). Latour distinguishes between different ways to bring 'worlds' into existence, such as religion, economics, or law. The philosophical backing is different criteria for truth. Personally, I do not fully endorse his

approach, but it connects with many other philosophical streams that, most generally, analyse 'social ontology'. One of the most concise approaches is Searle's distinction between 'mind dependent' and 'mind independent facts'. Well, 'mind dependent facts' are – facts. That means, they have the same ontological status as other 'facts' conventionally treated as such by the sciences. Turning to the discount model again, a core question is the ontological status of institutions. From the viewpoint of institutional economics and Searlian philosophy of institutions, the authors appear to be imprecise in treating institutions because they seem to suggest that a clear boundary can be drawn between CUL and MET. Granted, there is the overlap in the diagram, but what does that mean? I think the problems crystallize in the question how to deal with the economy: Is the economy a 'world' of its own? It is a very specific and very comprehensive institutional structure that creates 'realities' to which we need to adapt, in the eyes of many (the infamous TINA principle). Accordingly, some economists think that the discount rate should follow the market interest rate, as this is the only way to define a 'collective discount rate' apart from individual time preferences. Markets are real, everything else is 'subjective'. This 'objectifies' markets just in the Searlian sense. Indeed: Which other way do we have to generate a 'country' discount rate in an empirically meaningful way? The authors introduce this with a sleight of hand, but this is a very strong ontological projection! I think the discount model cannot simply take a 'country discount rate' for granted – unless that would be the discount rate that governments apply in their policy making framework. In my own work, following dystopian statements by experts such as Pyndick (Pyndick, R. S. (2013): Climate Change Policy: What Do the Models Tell Us? Journal of Economic Literature LI (3), 860-872), the discount rate is the central parameter that manifests the mutual irreducibility and incommensurability of economy and ecology, which becomes manifest in the methodological troubles of IAM. The authors refer to the fact that economic models always refer to monetary variables (p. 8) – that is what constitutes a 'world' in the Searlian or Latourian sense. Thus, I wonder whether a minimum requirement for building a discount model is to include a model of the economy that generates a reference interest rate as a 'social fact'. 3. I

can only hint at these issues here, but the central question arising from this is how to justify the assumption of an integrated 'World-Earth system'. I would bet on a 'Multiple Worlds-Earth ….' – system? Again, Latour has a strong point in rejecting the very notion of a 'system' as that would imply integration, coherence, and so on. In more practical terms, that leads us to consider the question why the authors did not follow a more traditional (hence probably outdated) approach in distinguishing between different 'systems', such as 'the economy', 'the society' and so on, which are not integrated, but may stay in fundamental tensions and contradictions with each other. Again, this is not merely a philosophical issue. In the Anthropocene literature, many critics point out that it is misleading to confront an abstract notion of 'human system' with 'Earth system' as this papers over the fact that the 'human system' is deeply fragmented and conflict ridden, and hence fundamentally politicized in a most general sense (as an exemplary work, see Bonneuil, Christophe and Jean-Baptiste Fressoz. 2017. The Shock of the Anthropocene: The Earth, History and Us. Verso London New York). This is the real challenge that the authors must meet: Catching this complexity in ONE model. Given these caveats, I think that a comprehensive approach would indeed be well advised to go back to ontological fundamentals. Mario Bunge's formal ontology seems still unsurpassed to me (Bunge, Mario (1979), Treatise on Basic Philosophy, Volume 4. Ontology II: A World of Systems, Dordrecht: Reidel). Perhaps this would allow for formalizing 'multiple worlds'. I am not sure how a model would look like. Perhaps it would be a set of modules that are only loosely integrated, with certain thresholds that block interaction below them, under normal circumstances. That means, for example, the 'economic world' would generate a discount rate that would be isolated from other worlds, until a catastrophe happens which triggers sudden spillovers. Modules would run separately for longer periods, until connections become activated that would generate sudden changes across the modules. The discount model might consist at least of two modules, one economic, the other cultural. The latter would include public opinion, value changes, and so on. I think this is what we observe in reality: Public opinion might shift towards de-carbonizing the economy, but the economy seems to move towards

sustaining it endogenously (think of the continuous process of postponing depletion of fossil fuel reserves) (see Covert, T., M. Greenstone und C. R. Knittel (2016): Will We Ever Stop Using Fossil Fuels? Journal of Economic Perspectives, 30(1), 117-38). Then, everyone is shocked learning that Germany even increases $CO_2$ emissions! 4. Another issue is the principles how to build a taxonomy. The authors work with causal concepts, such as 'mechanisms' and 'feedback loops'. This is a basic methodological requirement, for sure. But it implies that the taxonomy must be based on theories and hypotheses about causality. This creates a very, very high benchmark. Taxonomical work is often more modest, such as biological taxonomy that is based on notions of descent, similarity and so on, without classifying patterns of underlying causalities. This is one advantage of working with old fashioned systems categories: They offer simple criteria of building taxonomic classes, such as treating the economy as a system in which money and monetary values are coordinating media, and then subdividing this in other systems, such as sectoral, regional or developmental. That would imply for the discount rate, for example, that one would minimally distinguish between 'North' and 'South' as economic systems which have different reference rates (maybe North can afford to be more 'patient', South not). But if one refers to causality, every single example given by the authors is a Pandora's box out of which theoretical controversies and cross-disciplinary battles emerge (in fact, the authors often mention that many disciplines deal with the various mechanism that they subsume under their categories). Just take one: there is the ENV-CUL taxon, and cultural identity as 'sense of place' is mentioned. I assume that many cultural geographers would immediately protest. Does that mean that there is environmental determinism regarding culture? I do not think that the authors believe that, and after all, there is also the CUL-ENV taxon. But how can we deal with this in building a model for one cultural element, 'sense of place' (which certainly is important in the context of modelling migration, induced by climate change)? In this single example, very complicated theoretical issues regarding causality, its direction, and divergences between different disciplines are involved. Which of the competing positions in the literature would be selected to model this single element? Clearly, that would directly affect the taxonomy. In other words, the taxonomy is built on a minefield of theoretical and cross-disciplinary controversies. The difficulties multiply if one considers larger complexes, such as 'the economy'. I appreciate very much that the authors recognize complex feedbacks such as the cultural shaping of preferences, or their technological determinants. But all this is object of deep controversies between the different disciplines. 'The economy' looks very different in the eyes of the economist or the sociologist. Which position should be included in the 'World-Earth-System' model? Even the most basic assignments to taxa would be affected by this. For example, many sociologists would assume that 'the economy' belongs to the socio-cultural taxon, and that repercussions on the other taxa work via technology. My comment is already very long, so I stop here. I am afraid that my comments appear overly destructive, but this is not my intention. I hope that the authors take it as creative stimulus. Yet, my general point is serious: If the taxonomy is about causal patterns, it needs to be built on theories. The authors assign various elements to different taxa with levity, without considering that all this is subject to many competing and often contradictory theories across the entire universe of disciplines, not only the sciences, but even the humanities. Evidently, their model must also be a model of cross-disciplinary relations, to avoid unjustified essentialization and hypostasis of assumed mechanisms, loops etc. This is, well, the ultimate 'Theory of Everything' coming along in the disguise of a model? I hope that we will continue with this discussion and hope to learn from the authors' response, as well as from the comments of other reviewers.

---

## Editor Comment (EC1) · J. Dyke (Editor) · 10 Jul 2018

This review has been posted on behalf of Reviewer 2.

====

This manuscript addresses a highly relevant topic on how to foster the integration of the social dimension in mathematical and computer models up to the planetary scale. A specific focus is set on how to take into account the relevant social-ecological feed-backs. It fits well in the scope of the journal. The proposed taxonomy (including the

types of interactions between the subsystems) takes adequately the existing range of global environmental change models into account and classifies them. In addition it enables to reveal subsystems and interactions which are important but underrepresented in global scale models. Therefore this paper can serve as a fruitful starting point for a more structured approach to guide the future development for such types of highly needed models. However I see major points which have to be addressed in a revised version of this manuscript:

1. The three guidelines in constructing the taxonomy are partly not well explained, e.g. what do you mean by "compactness". In addition have these guidelines actually been tested? I ask that, because they are afterwards only mentioned in the conclusion section again.

2. The type of content of the subsections 3.1. – 3.9 is differing. I would suggest to offer results in each section for the same set of questions, such as: examples, what is reached, what are open challenges, at which levels are the current models prevalent (global yes or no).

3. Make more explicit what the added value of the presented example model for this paper is. Is it just to illustrate how a simple model could look like which includes interactions from almost all classes of the subsystems of the taxonomic scheme?

4. The text includes a number of assumptions where references which underpin the respective statement are missing (such as P2L21, P6L17ff), more examples below. Please add references.

In addition I have further comments for the different sections:

Abstract: - The term "World-Earth model" is not familiar to the readers. It is only explained later in the introduction. Perhaps a short explanation in the abstract could be helpful. The same holds for the term "higher-order taxonomies". Introduction:

- P3L5-6: The second half of the sentence is not comprehensible to me.

[Figure]

- P3L14: I think a number of ecologists (e.g. behavioural ecologists) would doubt that "laws of nature" govern for instance how animals behave. I understand the point you want to make. Perhaps add a footnote which points this out shortly.

- P3L27: delete "computer"? According to L24/L25 it holds for both types of models: mathematical and computer simulation models.

- P4L11: I suggest to give a reference to the term "safe and just operating space for humanity".

- P5L2: The term "mechanism" is defined, but never used in the paper. Section 2:

- P6L17: "Growing reliance on model-based insights for global decision making": please give references for that.

- P7L15: Add a reference for "deep future" studies. Where comes the term from?

- P7L24: You describe "these models have tended to be small-scale, context-specific...". There are counterexamples (such as LPJ or large scale forest models). Hence, I suggest to mention these examples.

- P8L7: "Our approach can bring much-needed clarity and transparency about the role of such models..." – this statement is not substantiated in the paper to my point of view. Section 3:

- P9L29: It could be helpful to add a short paragraph in this paper how the taxonomy have proven to be helpful building the copan:CORE framework.

- Figure 2: I find it not self-explanatory, why certain interaction examples are assigned to a certain category, for instance why "needs" is in MET CUL. I suggest to explain at least those in the text.

- P12L23: Make sure that all abbreviations in the text are once spelled out and the abbreviation is written in parentheses (e.g. ESM or later BAU)

- P13L11 Use the same spelling in the whole manuscript: Noösphere, Noosphere

- P14L19: "may be subsystems from ENV, MET or CUL". To my understanding "or CUL" has to be deleted, since the relationship CUL- CUL feedback loops seems not to be included.

- P14L24: Give examples what you mean by 3-loops. For me it is unclear how you get 11 taxa.

Section 4: - P18 Figure 4: Add the parameter values for which the graphs are generated, in particular for the adjustment of time preferences. Reason: I would like to see all necessary information that the reader may recalculate the results.

- Twice in the caption of Figure 4 "S" instead of "C" is used.

Section 5: - P19L21: I do not see how the provided framework may be helpful as "a blueprint for constructing alternative taxonomies." This statement needs to be substantiated.

Appendix: - P27L3: Eqs 1 and 2.

- P27L27: You call both gamma and s(c) damage factor. I suggest to be more precise and to not use the same name for (slightly) different elements of the equations.

- P27L28: Be more precise throughout the Appendix and add the (t) with the C

- P28L13: Check the size of the parentheses for better understanding: I have impression that left of G it should be a large parenthesis

---

## Author Comment (AC1) · 17 Jun 2019

**Response letter "Taxonomy paper"**

We thank the two reviewers for their thoughtful, detailed and very helpful comments on the presented framework and the presented illustrative DISCOUNT model. In our point-to-point response below, we reflect on their thoughts and propose according changes to be made to the manuscript in a next step of revision.

Our responses are highlighted in italics. References referred to in the responses are listed at the bottom of the document.

The reviews and our responses refer to the following discussion paper:

> Donges, J. F., Lucht, W., Heitzig, J., Barfuss, W., Cornell, S. E., Lade, S. J., and Schlüter, M.: Taxonomies for structuring models for World-Earth system analysis of the Anthropocene: subsystems, their interactions and social-ecological feedback loops, *Earth Syst. Dynam. Discuss.*, https://doi.org/10.5194/esd-2018-27, in review, 2018.

**Reviewer 1 (C. Herrmann-Pillath)**

This is a timely and much-needed effort at creating more realistic and inclusive/ integrative models of the Earth system that meet the challenge to model the complexities of human action in various systemic contexts (such as economy, society and so on). It is far too easy to criticize such endeavours because they necessarily offer many points of attack, being the first steps in this direction. This would be unfair. In my review, I try to find a proper balance between necessary criticism on the one hand, and endorsement and encouragement on the other hand. I am writing from the position of an economist working in the field of evolutionary, institutional and ecological economics (hence, clearly 'heterodox') who always injects a strong dose of philosophy in his work. That is, I am an outsider to the modelling community in the Earth and Climate Sciences. I will always combine general observations with specific comments on the discount model. Yet, my focus is on principled issues, hoping that this might provide different perspectives than those within the modelling community. The authors explicitly recognize the philosophical dimensions of their work; hence I start out from there, and I congratulate them for showing the grit to use the notion of 'World-Earth system'
(p. 2). However, this immediately raises many philosophical questions.

> *Outside scrutiny is very much welcomed! The rationale for the paper is to support transdisciplinary knowledge integration, so the broader the scrutiny and critique, the more robust the usability for our target communities.*

1. In which sense can we separate 'Earth' from 'World'? Although we must assume that there is an 'Earth' interacting with 'World', the 'Earth' is only accessible via the sciences which construct it as a 'World'. That means, we must recognize the fundamental fact of our incomplete knowledge about 'Earth'. In other words, **a complete model** of the 'World-Earth system' must be reflexive in the sense of including the Earth Sciences (and others) as endogenous generator of our scientific conceptions of 'Earth'. There are direct implications for

the subsystems: For example, our knowledge about the biophysical mechanisms will impact on public perceptions of policy issues, and so on (which the authors aim at modelling explicitly, indeed). These will feedback on funding Earth Systems sciences, and hence will determine how 'Earth' will appear to us in future times. I think that science is an essential part of the 'social-cultural taxon' and cannot just be imagined as being exogenous. But if it is endogenous, 'Earth' is endogenous.

> *We appreciate these comments and critiques because they help us to refine our transdisciplinary work, which increasingly extends beyond the Earth and Climate sciences modelling community. What we do is not about representing an ideal or "complete" world view, but is aimed at making things explicit that have so far been excluded from Earth system schema. We are proposing a taxonomy for models-in-use - an applied epistemological standpoint - not a taxonomy for worldviews - ontologies, even though of course the ontological stances of modellers have consequences for the "real world".*

> *However, the feedbacks discussed by the reviewer are indeed important and are, for example, very much in the core of ideas that have been expressed on the nature of World-Earth in the Anthropocene as Gaia 2.0 (Lenton and Latour, 2018) or the emergence of a global subject (Schellnhuber, 1998, 1999). Scrutinizing the essence of such feedbacks using simulation models is clearly an ambition of the type of the science we describe in the manuscript, but needs to be approached step-by-step starting with reduced and stylized models.*

Looking at the discount model, this seems implicit to the role of the parameters p and q. Although these are essential for driving the specific model dynamic in determining the probability of switching discount rates, they are not explained in any way. To be frank, I have difficulties in understanding their meaning. What exactly is 'curiosity' of a country? What is 'myopic rationality'? Many people would just think that the latter means, well, steep discounting of the future. Another definition would be the length of the time horizon, which differs from discount rates applied within that horizon. Thus, besides from the need to give precise definitions these parameters and explain how they could be measured, …

> *We appreciate these detailed thoughts about our illustrative model and its parameters. We believe, however, that there are some misunderstandings both regarding our aims in presenting this model and also regarding the pair of parameters you mention, p0 (we take it, this is the one you refer to as p) and q. In short, we want to show properties of the model in a way that is accessible to readers from whatever field of 'World' or 'Earth' system modelling they come from, so we want to use descriptive rather than specialist/technical names for key parameters.*

> *In our illustration, we were interested mainly in the dynamics of social learning, so we comment briefly on the influence of parameter l in the model. We call l the "learning rate" since it is the rate at which countries update their discount factor by social learning. They do this by comparing their welfare with others', and then switch to the other discount factor with some probability. The effect of varying this parameter is shown in Fig. 4, with l being small in the top panel and large in the bottom panel. As*

*detailed in the Appendix, the illustrative model has about a dozen exogenous governing parameters, of which p0 and q are far from being the most important ones. As can be seen in Eq. 6, p0 and q influence the frequency at which discount rates are switched. As the reviewer notes, these two parameters control the sigmoid-shaped dependency of the switching probability at each of the updates that occur at rate l. p0 is the probability of switching in the case that both countries' welfares are equal. We named p0 a "curiosity" parameter because a switch of this kind can happen without any social learning by imitation. This parameter can be interpreted as a country's "look-see" exploration of a different discount factor without expecting a welfare increase. We now make this clearer in the manuscript text:*

> *Page 29, line 16: "p0 can be interpreted as a measure of a country's curiosity-driven exploration of a different discount factor without expecting a welfare increase. The larger p0, the more frequent switches will occur, but in both directions between the two candidate discount rates, mainly generating more variance and fluctuations that can be seen as a form of "noise"."*

*The parameter q is the steepness of the probability curve at the point where both countries' welfares are equal. In other words, it is the marginal probability at this point w.r.t. welfare differences. In the extreme case where q goes to infinity, the probability of switching is either one (if the other country has higher welfare) or zero (otherwise). One could interpret this as a "rational" but "myopic" switching behaviour, because the agent does not anticipate the future in their decisions. We now make this clearer in the manuscript text:*

> *Page 29, line 17: "q can be interpreted as a measure of a country's rationality, because the probability of switching to the other country's discount rate is higher if the other country has higher welfare (and zero if that is not the case) - but it is a myopic rationality, because the agent only takes its present welfare into account. The larger q, the faster discount factors will converge to the one currently generating the largest welfare."*

*In our terminology, myopia vs. farsightedness (the parameter q) refers to the agent's anticipation of future welfare, while patience vs impatience (the application of a large value for δ) refers to whether an agent cares about and values future welfare in their decisions. So, in our model, a myopic country may switch from patient to impatient because at that moment, the impatient countries seem to fare better and the country does not anticipate (due to its myopia) that this leads to a trajectory on which welfare will later be smaller for impatient than for patient countries.*

*The influences of p0 and q on the speed of the learning dynamics are more indirect and subtle than the influence of the learning rate l. The reviewer is right to observe that the equations given in the Appendix, though complete, are not sufficient to demonstrate this to the reader without access to model runs. Given that our intention in showing this stylized model is just to illustrate the taxonomy and to demonstrate a small point - that the speed of $CUL \to CUL$ learning dynamics is important for coevolutionary model outcomes - we hope that the clearer explanation of our lay-terms*

*for parameters in our World-Earth illustration model and the clarifications about the behaviour of the parameters in the Appendix text and will suffice.*

… I suggest that they refer to the endogeneity of science. Just consider Trumpian America: the recent issue of 'The Economist' has an article about 'Swamp Science' at the EPA. This is actively reducing 'curiosity', it seems to me. In other words, I think that if one uses the 'World-Earth' duality, one faces the challenge of treating human knowledge about 'Earth' as endogenous. There is no external stand-point of the model-builder. Another excellent example for this problem is the treatment of 'damage' and 'welfare' in the discount model. It seems that the authors think that there is an objective measure of welfare and damage. But we know that this is one of the most difficult and disturbing aspects of IAM, namely that the damage function is endogenous and depends on the discount rate. That is why some economists now even take the very radical step to build their models without damage function (Llavador, H.,J. E. Roemer und J. Silvestre (2015): Sustainability for a Warming Planet. Cambridge University Press). But behind this is the simple, but deeply philosophical problem that neither welfare nor damage can be assessed from the standpoint of an external observer. Apparently, the authors are aware of this, as on p. 17 we find the expression:
s(C) (ENV → MET → CUL). But that implies that social learning may not only happen via imitation of discount rates, but also via diffusion of valuations, or, 'worldviews'.

*We appreciate these concrete examples, and share much of the reviewer's concern about the state of science in society. We now indicate the endogeneity of science in the abstract:*

> *Page 1, line 11: "(ii) socio-cultural, dominated by processes of human behaviour, decision making and collective social dynamics (e.g., politics, institutions, social networks, and even science itself)*

*We have also added a reference, Yearworth and Cornell 2016, that discusses issues around the shifting role and stance of the scientist/model builder in different sustainability contexts:*

> *Page 8, line 22: Notably, the CUL taxon also includes processes of digital transformation and artificial intelligence that increasingly restructure and shape the socio-cultural sphere of human societies. It also provides a locus for debating the challenge of reflexiveness in science, especially in fields where modelling plays a vital role in shaping knowledge and action (Yearworth and Cornell 2016). For instance, future World-Earth modelling will have to grapple with ways to recognize Earth system science as an endogenous generator of scientific conceptions of 'Earth'.*

*We are confident that our taxonomy can be useful in diagnosing the shortcomings of real-world science and models-in-use - as in the reviewer's examples given above. But fundamentally, our taxonomy is intended to provide a system for categorizing and classifying the structure of models, their subsystems, and indeed also their couplings and hybridizations as efforts for improved representation and understanding of "whole"*

*Earth system processes and phenomena. It is not a recipe for a new model, nor is it a taxonomy of issues in the real world.*

There is no external stand-point of the model-builder:

*This is a very good point. One reason for this taxonomy is that some new model developments articulate this point explicitly, while many others do not. And also some new model developments that are being coupled to existing "Earth" models involve an internal positioning of the model builders – e.g., participatory modelling, etc. By making this taxonomy, we develop some initial tools and terminologies for systematically challenging model builders and model users to be clear about their social/cultural and perhaps also their epistemological/axiological standpoints.*

2. The next foundational question is whether there is 'ONE World', which seems implicated by putting both World and Earth together into one 'system'. Obviously, this is not just annoying sophism: There are many philosophers who cast doubt on that (just mention the German philosopher Markus Gabriel with the provocative book title 'Why there is no World' 'Warum es die Welt nicht gibt'). The serious argument behind it ties up with the previous: Science constructs worlds, and there is no necessity that these are just 'one'. This is even more evident if we consider human worlds: Actually, it is the CUL taxon that creates the 'worlds'. I think that it would be most helpful for the authors to look at Bruno Latour's recent work on Gaia and the more general work on 'modes of existence' (see [http://modesofexistence.org/)](http://modesofexistence.org/). Latour distinguishes between different ways to bring 'worlds' into existence, such as religion, economics, or law. The philosophical backing is different criteria for truth. Personally, I do not fully endorse his approach, but it connects with many other philosophical streams that, most generally, analyse 'social ontology'. One of the most concise approaches is Searle's distinction between 'mind dependent' and 'mind independent facts'. Well, 'mind dependent facts' are – facts. That means, they have the same ontological status as other 'facts' conventionally treated as such by the sciences.

*We really appreciate this opportunity for cross-disciplinary critique. We would argue that we do not need to think there is "one world" (an ontological position) in order to want to combine models of different worlds (an effort at handling some specific challenges arising from epistemological plurality) in order to seek better scientific understanding of aspects of the (real!) world. And yes, we agree with the more general point that models are things in the world that may have causal power (agency in Latour's terms).*

*Tackling the epistemic point first – in short, each model of the system contains one version of "world" - and the modeller knows this is a simplified/stylized representation with a partial viewpoint. Different models can allow for the comparison of different versions of "world". Later, some "metamodel" could allow for several versions of "world" in one model, but this only makes sense if they interact, e.g. in that different agents are guided by different conceptions of "world".*

*This returns us to the ontic challenge - in the context of the models we want to "taxonomise",  one can say that what we call "world" is the modeler's view of the world, while the agents' views of the world are likely to be part of the CUL-related attributes*

*of the agents. But that is not necessarily the only option. Some researchers are applying Latour's ideas on agency to non-human things (like water in irrigation systems), which our taxonomy would flag up as requiring specific attention to sub-system interactions (for instance, perhaps modelling different CUL-MET behaviours than human agents).*

Turning to the discount model again, a core question is the ontological status of institutions. From the viewpoint of institutional economics and Searlian philosophy of institutions, the authors appear to be imprecise in treating institutions because they seem to suggest that a clear boundary can be drawn between CUL and MET.

*We did not mean to suggest a rigid boundary. Although we only talk explicitly about institutions in the CUL sections, we mention kinds of institution (e.g., economic and technological systems) in MET. In fact, as in all taxonomies, it is often not clear where to put a given process without having information about the underlying basis. Still, we believe a distinction between rather socio-cultural processes (let's say mind-mediated) and rather socio-metabolic processes (expressed as material flows) is helpful in modeling,nwhich is what this paper is about. Pragmatic assignments of boundary cases to either CUL or MET are typically not harmful. In the discount model, the relevant institutions are represented in a very stylized way in two components, (i) the game-theoretic submodel and (ii) the social learning submodel. The first specifies which domestic emissions will arise at every point in time as a kind of "immediate" Nash equilibrium where each country is assumed to act as one consistent player, as is standard in the literature on international environmental agreements that this submodel is taken from. This of course means that we implicitly assume strong decision-making institutions within each country. We hence interpret this submodel to basically belong to the CUL taxon, where part of its inputs (namely the evaluation of the environmental and socio-metabolic states) come from the ENV and MET taxon, and the resulting emissions caps have effects in the MET taxon. The second submodel is also adapted from standard models, this time from the modeling literature on social learning. Implicitly it assumes some global monitoring and communication institutions that allows "observing" other countries' discount factors and evaluations in some way. We interpret this to be a mainly socio-cultural process as well. Hence the reviewer is right in that we treat institutions "imprecisely" in the same way that the game-theoretic literature on international environmental agreements and the modeling literature on social learning do.*

Granted, there is the overlap in the diagram, but what does that mean? I think the problems crystallize in the question how to deal with the economy:
Is the economy a 'world' of its own? It is a very specific and very comprehensive institutional structure that creates 'realities' to which we need to adapt, in the eyes of many (the infamous TINA principle).

*Here we reiterate that our aim is to develop a taxonomy to help classify and structure modelling approaches. We are not prescribing how global models should be configured or what they should contain, nor are we critiquing the power of some global models in shaping contemporary economic realities. In sections 2.2 and 2.3, we indicate some of the diversity of ways of dealing with the economy, and in section 2.4 we identify*

*some research areas that deal with adaptive change and transformation. So we think our taxonomic approach can help in tracing how and where "the economy" (which also is not universal/monolithic!) actually features in contemporary analysis of Earth system dynamics sensu lato.*

Accordingly, some economists think that the discount rate should follow the market interest rate, as this is the only way to define a 'collective discount rate' apart from individual time preferences.

*This might be true, but we tend to disagree with those economists then and rather follow the illustrious group around the late Kenneth Arrow who, when asked by the EPA what discount rate a country should use for long-term projects like fighting climate change, basically said that this is a normative choice: "Many of us regard the Ramsey approach to discounting, which underlies the theory of cost-benefit analysis, as a normative approach. This implies that its parameters should reflect how society values consumption by individuals at different points in time; i.e., that δ and η should reflect social values." (Arrow, Kenneth, et al. "How should benefits and costs be discounted in an intergenerational context? The views of an expert panel." (2013)). In our model, we make the specific and certainly highly debatable assumption that this choice is made by social learning between countries, mainly to illustrate what effects such "non-economic", "social" dynamics could have.*

Markets are real, everything else is 'subjective'. This 'objectifies' markets just in the Searlian sense. Indeed: Which other way do we have to generate a 'country' discount rate in an empirically meaningful way? The authors introduce this with a sleight of hand, but this is a very strong ontological projection! I think the discount model cannot simply take a 'country discount rate' for granted – unless that would be the discount rate that governments apply in their policy making framework.

*You are of course right, but the model's discount rate is indeed used in the way you suggested, as the one the government uses to decide their domestic emissions caps.*

In my own work, following dystopian statements by experts such as Pyndick (Pyndick, R. S. (2013): Climate Change Policy: What Do the Models Tell Us? Journal of Economic Literature LI (3), 860-872), the discount rate is the central parameter that manifests the mutual irreducibility and incommensurability of economy and ecology, which becomes manifest in the methodological troubles of IAM. The authors refer to the fact that economic models always refer to monetary variables (p. 8) – that is what constitutes a 'world' in the Searlian or Latourian sense. Thus, I wonder whether a minimum requirement for building a discount model is to include a model of the economy that generates a reference interest rate as a 'social fact'.

*Our aim in presenting this conceptual illustrative discount model is precisely that - just to show that discount rates may differ between countries and over time, contrary to the predominant assumptions of IAMs. Building a detailed model of financial markets would only be only necessary for answering research questions related to aspects of those markets.*

3. I can only hint at these issues here, but the central question arising from this is how to justify the assumption of an integrated 'World-Earth system'. I would bet on a 'Multiple Worlds-Earth....' – system? Again, Latour has a strong point in rejecting the very notion of a 'system' as that would imply integration, coherence, and so on. In more practical terms, that leads us to consider the question why the authors did not follow a more traditional (hence probably outdated) approach in distinguishing between different 'systems', such as 'the economy', 'the society' and so on, which are not integrated, but may stay in fundamental tensions and contradictions with each other. Again, this is not merely a philosophical issue. In the Anthropocene literature, many critics point out that it is misleading to confront an abstract notion of 'human system' with 'Earth system' as this papers over the fact that the 'human system' is deeply fragmented and conflict ridden, and hence fundamentally politicized in a most general sense (as an exemplary work, see Bonneuil, Christophe and Jean-Baptiste Fressoz. 2017. The Shock of the Anthropocene: The Earth, History and Us. Verso London New York). This is the real challenge that the authors must meet: Catching this complexity in ONE model. Given these caveats, I think that a comprehensive approach would indeed be well advised to go back to ontological fundamentals. Mario Bunge's formal ontology seems still unsurpassed to me (Bunge, Mario (1979), Treatise on Basic Philosophy, Volume 4. Ontology II: A World of Systems, Dordrecht: Reidel). Perhaps this would allow for formalizing 'multiple worlds'. I am not sure how a model would look like. Perhaps it would be a set of modules that are only loosely integrated, with certain thresholds that block interaction below them, under normal circumstances. That means, for example, the 'economic world' would generate a discount rate that would be isolated from other worlds, until a catastrophe happens which triggers sudden spillovers. Modules would run separately for longer periods, until connections become activated that would generate sudden changes across the modules. The discount model might consist at least of two modules, one economic, the other cultural. The latter would include public opinion, value changes, and so on. I think this is what we observe in reality: Public opinion might shift towards de-carbonizing the economy, but the economy seems to move towards sustaining it endogenously (think of the continuous process of postponing depletion of fossil fuel reserves) (see Covert, T., M. Greenstone und C. R. Knittel (2016): Will We Ever Stop Using Fossil Fuels? Journal of Economic Perspectives, 30(1), 117-38). Then, everyone is shocked learning that Germany even increases CO2 emissions!

> *This comment raises several points that we condense into the following responses: "The central question arising from this is how to justify the assumption of an integrated 'World-Earth system'" – "This is the real challenge that the authors must meet: Catching this complexity in ONE model" – "Perhaps this would allow for formalizing 'multiple worlds'".*
>
> *Our focus here is on integrated models - and the epistemology does not presuppose the ontology of one integrated World-Earth system. There are many approaches to representing aspects of the complex real World, as we indicate in our discussions of subsystem interactions (section 3). The many different modelling efforts that our contributing communities are making are not competing for ultimate victory over each other - they are capturing different aspects of the complex world. We are actually arguing for epistemological pluralism - we are seeking to enable productive dialogue and interaction between this diversity of World modelling approaches and the biophysical Earth representations that we already have.*

*So our underlying research aim is not to make One Model (to rule them all...), but to highlight what a new family of models might be. We are focusing on the specific toolkit of simulation modelling. Our approach already allows us to disaggregate and examine the complex social worlds much more than traditional IAM approaches, enabling a more politicised view on societies and economies by setting concepts and contexts into taxa. In other strands of work, we are trying to represent multiple economies and diverse "systems" coexisting. Our taxonomy can permit "intersectionality" in this diverse analytical context, improving the transparency that we think is needed for model-use in real world decision contexts.*

4. Another issue is the principles how to build a taxonomy. The authors work with causal concepts, such as 'mechanisms' and 'feedback loops'. This is a basic methodological requirement, for sure. But it implies that the taxonomy must be based on theories and hypotheses about causality. This creates a very, very high benchmark. Taxonomical work is often more modest, such as biological taxonomy that is based on notions of descent, similarity and so on, without classifying patterns of underlying causalities. This is one advantage of working with old fashioned systems categories: They offer simple criteria of building taxonomic classes, such as treating the economy as a system in which money and monetary values are coordinating media, and then subdividing this in other systems, such as sectoral, regional or developmental. That would imply for the discount rate, for example, that one would minimally distinguish between 'North' and 'South' as economic systems which have different reference rates (maybe North can afford to be more 'patient', South not). But if one refers to causality, every single example given by the authors is a Pandora's box out of which theoretical controversies and cross-disciplinary battles emerge (in fact, the authors often mention that many disciplines deal with the various mechanism that they subsume under their categories). Just take one: there is the ENV-CUL taxon, and cultural identity as 'sense of place' is mentioned. I assume that many cultural geographers would immediately protest. Does that mean that there is environmental determinism regarding culture? I do not think that the authors believe that, and after all, there is also the CUL-ENV taxon. But how can we deal with this in building a model for one cultural element, 'sense of place' (which certainly is important in the context of modelling migration, induced by climate change)? In this single example, very complicated theoretical issues regarding causality, its direction, and divergences between different disciplines are involved. Which of the competing positions in the literature would be selected to model this single element? Clearly, that would directly affect the taxonomy. In other words, the taxonomy is built on a minefield of theoretical and cross-disciplinary controversies.

*We view this issue from a position within a modelling community using big, rather opaque "comprehensive" global models - and we certainly do encounter controversies and cross-disciplinary tensions, if not necessarily all-out-battles. In other words, we are in the minefield! Every new collaborative effort needs to face these theoretical debates, and this effort is helped when we are alert to their historic and ideological loadings. The global models we and our colleagues use have causal power in the world, so our effort here has a partial aim to illuminate the innards of these tools. But we do not agree that the taxonomy needs to have an a priori set of theories and hypotheses about causality. Causal narratives are our starting point because they are necessary for and explicitly encoded in simulation modelling - and our classification lets us interrogate them more systematically and exposes them explicitly, as the CUL-ENV/ENV-CUL example above shows.*

The difficulties multiply if one considers larger complexes, such as 'the economy'. I appreciate very much that the authors recognize complex feedbacks such as the cultural shaping of preferences, or their technological determinants. But all this is object of deep controversies between the different disciplines. 'The economy' looks very different in the eyes of the economist or the sociologist. Which position should be included in the 'World-Earth-System' model? Even the most basic assignments to taxa would be affected by this. For example, many sociologists would assume that 'the economy' belongs to the socio-cultural taxon, and that repercussions on the other taxa work via technology.

> *Nicely, there is no need at all to place "the economy" (whatever that is) into MET or CUL, since our taxonomy is about classifying individual processes in specific models. Thus the question of whether a certain process is "economic" or (otherwise) "social" or "cultural" will depend on how it is framed. An economist's model would probably place the real-world process of decision-making about childrens' education into the MET taxon, and model it as a rational market-driven process linked to resource productivity, while a sociologist's model would probably place the same real-world process into the CUL taxon and model it as being strongly influenced by traditions.*

> *The taxonomy approach means that things that were previously included in models as opaque and unquestioned systems can be unpacked and critically examined. Model users who were not the model builders would really benefit from knowing (to take the example above) whether the representation of an education decision process in a model is in MET or CUL.*

My comment is already very long, so I stop here. I am afraid that my comments appear overly destructive, but this is not my intention. I hope that the authors take it as creative stimulus. Yet, my general point is serious: If the taxonomy is about causal patterns, it needs to be built on theories. The authors assign various elements to different taxa with levity, without considering that all this is subject to many competing and often contradictory theories across the entire universe of disciplines, not only the sciences, but even the humanities. Evidently, their model must also be a model of cross-disciplinary relations, to avoid unjustified essentialization and hypostasis of assumed mechanisms, loops etc. This is, well, the ultimate 'Theory of Everything' coming along in the disguise of a model? I hope that we will continue with this discussion and hope to learn from the authors' response, as well as from the comments of other reviewers.

> *We do take these comments as creative stimulus! This thoughtful attention has opened discussions about our own philosophical and research-ethical positions. We hope that our engagement with these points in our response help to "normalise" practical philosophical discussion in Earth system modelling contexts.*

**Reviewer 2**

This manuscript addresses a highly relevant topic on how to foster the integration of the social dimension in mathematical and computer models up to the planetary scale. A specific focus is set on how to take into account the relevant social-ecological feedbacks. It fits well in the scope of the journal. The proposed taxonomy (including the types of interactions between the subsystems) takes adequately the existing range of global environmental change models into account and classifies them. In addition it enables to reveal subsystems and interactions which are important but underrepresented in global scale models. Therefore this paper can serve as a fruitful starting point for a more structured approach to guide the future development for such types of highly needed models. However I see major points which have to be addressed in a revised version of this manuscript:

1. The three guidelines in constructing the taxonomy are partly not well explained, e.g. what do you mean by "compactness". In addition, have these guidelines actually been tested? I ask that, because they are afterwards only mentioned in the conclusion section again.

> We will make sure to explain these guidelines more explicitly in the manuscript along the lines of the following considerations:

> Defining **compactness** - we want a "top-level" taxonomy with as few classifications as possible, covering the scope of co-evolutionary modelling research parsimoniously and in a self-containing way. A contrasting approach would perhaps have been a kind of family tree, where classifications can expand almost ad infinitum because there's no basis for limiting the number of offspring. The discount model is a kind of test of this guiding principle - the loops or flows between taxa do not make us need to rethink the whole structure.

> Defining **operative capacity** - we start from a very dominant existing divide between "natural sciences" (e.g., fits mostly ENV and parts of MET) versus "social sciences" (e.g., fits mostly CUL and parts of MET), so our taxonomy needs to be able to include things that fit into those existing classifications - we want to be able to draw on the wealth of previous modelling efforts – while expanding to be more inclusive on both those fronts as well as allowing more differentiation and permutations of approaches within and between and beyond the old natural/social sciences divide.

> Defining **compatibility** with existing research fields and modeling methodologies - this is linked to Reviewer 1's comments on philosophies. The taxonomy isn't defined with a single "ism" in mind. It clearly has realist foundations as the vast majority of modelling methodologies do, but it is (perhaps surprisingly) agnostic about how this relates to different epistemological stances. The reason we care about this guiding principle is that we want to explore how to bridge across currently very distinct modelling approaches as well as to trace how the techniques chosen/used can relate back to the theories, assumptions, and framings of the contributory disciplines. Collins and Evans' book Rethinking Expertise has some interesting discussions on the "translational" expertise that is needed to bridge disciplines, but stops short of thinking of how this kind of expertise exists outside the head and presence of the researcher: Our

*taxonomy both demonstrates and provides a way that we as a transdisciplinary community can learn from each other, not just from own experience.*

2. The type of content of the subsections 3.1. – 3.9 is differing. I would suggest to offer results in each section for the same set of questions, such as: examples, what is reached, what are open challenges, at which levels are the current models prevalent (global yes or no).

*Point well taken. We will restructure Section 3 for a more consistent presentation of the subsystem interaction taxa.*

3. Make more explicit what the added value of the presented example model for this paper is. Is it just to illustrate how a simple model could look like which includes interactions from almost all classes of the subsystems of the taxonomic scheme?

*This is indeed the main purpose since we feel that a crisp example such as this is needed to give the reader a feeling of where processes may be placed. At the same time, it also illustrates that even though in order to cover all classes of real-world processes that appear relevant in major global feedbacks one may have to combine quite different modeling traditions, this can still lead to macroscopic approximations that may be studied with established dynamical systems methodology. However, we also believe that the chosen model could be of interest in itself for some readers since to our knowledge it is the first published model which endogenizes the choice of discount factors used in climate policy.*

4. The text includes a number of assumptions where references which underpin the respective statement are missing (such as P2L21, P6L17ff), more examples below. Please add references.

*Thank you for spotting this, we will add references to underpin these statements.*

In addition I have further comments for the different sections:

Abstract:
- The term "World-Earth model" is not familiar to the readers. It is only explained later in the introduction. Perhaps a short explanation in the abstract could be helpful. The same holds for the term "higher-order taxonomies".

*Changes have been made to the abstract to clarify both these terms:*

> *add "World" to line 3 of abstract:*
> *New approaches to global modelling of the human World are needed to address these challenges. The current..."*

> *Also later: "World-Earth models capable of simulating the complex processes of the Anthropocene are currently not available. They will need to draw on and selectively integrate elements from all the existing modelling approaches..."*

> *To clarify "higher order taxonomies" in the abstract, split and expand the*

*sentence:*
*We show how higher-order taxonomies can be derived for classifying and describing the interactions between two or more subsystems. This then allows us to highlight the kinds of social-ecological feedback loops where new modelling efforts need to be directed.*

*On page 10 line 26, we add a further explanatory note:*

*"Our taxonomic approach can be extended to higher-order taxonomies that allow the classification of feedback loops and more complex interaction networks between subsystems. In Sect. 3.10, we briefly discuss a possible example."*

*At risk of sounding Latourian, the "World-Earth" framing signifies a growing attention (in many research fields) to the human world and biophysical earth as deeply connected, or a dynamic hybrid. Many very influential models were designed for World and Earth as separate domains. We are interested in the development of coevolutionary model toolkits that allow us to represent and explore this mutually entwined view.*

Introduction:
- P3L5-6: The second half of the sentence is not comprehensible to me.

*"the characteristics and interactions of social and biophysical subsystems are often not explicit to each other, if they are recognised at all" - changed to "the basis of the characteristics and interactions of social and biophysical subsystems are not explicit… Often, the interactions between these subsystems are not recognised at all.*

- P3L14: I think a number of ecologists (e.g. behavioural ecologists) would doubt that "laws of nature" govern for instance how animals behave. I understand the point you want to make. Perhaps add a footnote which points this out shortly.

*""natural laws" of physics, chemistry or ecology (e.g., atmosphere and ocean as governed by the laws of fluid- and thermodynamics)" - changed to "the "natural laws" and generalizable principles of physics, chemistry and (to some extent at least) ecology (for example, atmosphere and ocean circulation as governed by the physical laws of fluid and thermodynamics)"*

- P3L27: delete "computer"? According to L24/L25 it holds for both types of models: mathematical and computer simulation models.

*We will adjust the sentence accordingly.*

- P4L11: I suggest to give a reference to the term "safe and just operating space for Humanity".

*We will add references to Rockström 2009, Raworth 2012 and Dearing 2014 here.*

- P5L2: The term "mechanism" is defined, but never used in the paper.

*:-) Interesting observation. In several places, we refer to feedbacks, which are an important kind of mechanism in World-Earth system analysis. We have inserted the missing word mechanism in places where it clarifies our points about modelling feedbacks:*
  - *Page 3 line 7; page 3 line 30; page 4 line 27,*

Section 2:
- P6L17: "Growing reliance on model-based insights for global decision making": please give references for that.

*We will add references to:*

- *Calder M et al. 2018 Computational modelling for decision-making: where, why, what, who and how. R. Soc. opensci .5: 172096. http://dx.doi.org/10.1098/rsos.172096*
- *National Research Council. 2007. Models in Environmental Regulatory Decision Making. Washington, DC: The National Academies Press. https://doi.org/10.17226/11972.*
- *Sort of also a theme in: Rounsevell, M. D. A., Arneth, A., Alexander, P., Brown, D. G., de Noblet-Ducoudré, N., Ellis, E., Finnigan, J., Galvin, K., Grigg, N., Harman, I., Lennox, J., Magliocca, N., Parker, D., O'Neill, B. C., Verburg, P. H., and Young, O.: Towards decision-based global land use models for improved understanding of the Earth system, Earth Syst. Dynam., 5, 117-137, https://doi.org/10.5194/esd-5-117-2014, 2014.  - although they are focused on improving models by including decision processes more than improving decision processes that already rely on their models.*

- P7L15: Add a reference for "deep future" studies. Where comes the term from?

*We now add a couple of references to clarify our choice of this term, an analogy with "deep past". Curt Stager used the term in a popular science book - Deep Future: The Next 100,000 Years of Life on Earth - but we were not thinking of that text as such.*

*"Furthermore, as it becomes clearer that palaeoclimate models designed for study of the deep past can play a vital role in "deep future" studies of human-controlled processes in the Anthropocene (e.g., Zeebe R E and Zachos J C. 2013 Long-term legacy of massive carbon input to the Earth system: Anthropocene versus Eocene. Phil Trans R Soc A 371: 20120006.http://dx.doi.org/10.1098/rsta.2012.0006, also refs in Steffen et al 2018 Hothouse/Stabilized Earth paper)  …"*

- P7L24: You describe "these models have tended to be small-scale, context-specific ...". There are counterexamples (such as LPJ or large scale forest models). Hence, I suggest to mention these examples.

*Yes, we now mention them. We clarify our intended distinction of models of ecological dynamics as such (i.e., interactions between living organisms) versus models representing the physical dynamics of ecological processes and structures.*

- P8L7: "Our approach can bring much-needed clarity and transparency about the role

of such models ..." – this statement is not substantiated in the paper to my point of view.

> *Edited to say "We suggest that our approach can bring much-needed clarity and transparency about the role of such models in understanding the World-Earth system (cf van Vuuren et al 2016)".*

Section 3:
- P9L29: It could be helpful to add a short paragraph in this paper how the taxonomy have proven to be helpful building the copan:CORE framework.

> *Good suggestion, we will do that and explain how the CORE framework has been constructed following the taxonomy very closely.*

- Figure 2: I find it not self-explanatory, why certain interaction examples are assigned to a certain category, for instance why "needs" is in MET CUL. I suggest to explain at least those in the text.

> *A brief addition on material needs has been added to Section 3.6 ...*

- P12L23: Make sure that all abbreviations in the text are once spelled out and the abbreviation is written in parentheses (e.g. ESM or later BAU).

> *Thank you for pointing this out. We will check the text accordingly.*

- P13L11 Use the same spelling in the whole manuscript: Noösphere, Noosphere

> *We will make the spelling consistent.*

- P14L19: "may be subsystems from ENV, MET or CUL". To my understanding "or CUL" has to be deleted, since the relationship CUL- CUL feedback loops seems not to be Included.

> *We will check this point and correct if necessary.*

- P14L24: Give examples what you mean by 3-loops. For me it is unclear how you get 11 taxa.

> *3-loops are feedbacks that involve three different subsystems. We will expand on this point, give an example and explain how the 11 is arrived at (simply by combinatorics).*

Section 4:
- P18 Figure 4: Add the parameter values for which the graphs are generated, in particular for the adjustment of time preferences. Reason: I would like to see all necessary information that the reader may recalculate the results.

> *We have added all values [E0=1.6, c=1, r=0.45, l=0.2(top) or 1.3(bottom), gamma=1.1, mu=2, sigma=1, beta=0.1, alpha=0.5, G=2, N=50, p0=0.5, q=3)*

- Twice in the caption of Figure 4 "S" instead of "C" is used.

*We have fixed this.*

Section 5:
- P19L21: I do not see how the provided framework may be helpful as "a blueprint for constructing alternative taxonomies." This statement needs to be substantiated.

*We will expand on that point. The idea is that, for example, researchers may wish to develop their own taxonomies that have more detail in some aspects and less in others. E.g., one might wish to have a taxonomy otherwise similar to ours that distinguishes geophysical subsystems such as the atmosphere from ecological subsystems such as forests.*

Appendix:
- P27L3: Eqs 1 and 2.

*We have fixed this.*

- P27L27: You call both gamma and s(c) damage factor. I suggest to be more precise and to not use the same name for (slightly) different elements of the equations.

*We have fixed this.*

- P27L28: Be more precise throughout the Appendix and add the (t) with the C

*We chose to stick to just writing C in many places since it is clear from the exposition that this is one of the two dynamic variables that depend on t and it is common practise to avoid the explicit (t) in differential equations in natural sciences.*

- P28L13: Check the size of the parentheses for better understanding: I have impression that left of G it should be a large parenthesis

*Thank you for spotting this, actually there was a closing parenthesis missing, which is now fixed.*

**References**

Dearing, J. A., Wang, R., Zhang, K., Dyke, J. G., Haberl, H., Hossain, M. S., ... & Carstensen, J. (2014). Safe and just operating spaces for regional social-ecological systems. *Global Environmental Change*, *28*, 227-238.

Lenton, T. M., & Latour, B. (2018). Gaia 2.0. *Science*, *361*(6407), 1066-1068.

Raworth, K. (2012). A safe and just space for humanity: can we live within the doughnut. *Oxfam Policy and Practice: Climate Change and Resilience*, *8*(1), 1-26.

Rockström, J., Steffen, W., Noone, K., Persson, Å., Chapin III, F. S., Lambin, E. F., ... & Nykvist, B. (2009). A safe operating space for humanity. *Nature*, *461*(7263), 472.

Schellnhuber, H. J. (1998). Discourse: Earth System analysis—The scope of the challenge. In *Earth System Analysis* (pp. 3-195). Springer, Berlin, Heidelberg.

Schellnhuber, H. J. (1999). 'Earth system' analysis and the second Copernican revolution. *Nature*, *402*(6761supp), C19.

van Vuuren, D. P., Lucas, P. L., Häyhä, T., Cornell, S. E., & Stafford-Smith, M. (2016). Horses for courses: analytical tools to explore planetary boundaries. *Earth System Dynamics*, 7(1), 267-279.

Yearworth, M., & Cornell, S. E. (2016). Contested modelling: a critical examination of expert modelling in sustainability. Systems Research and Behavioral Science, 33(1), 45-63. https://doi.org/10.1002/sres.2315

---

## Referee Report (RR1)

The World Is Not Enough! On the Limits of 'Grand Modelling'

Carsten Herrmann-Pillath

The authors present a laudable effort to enrich existing models of Earth system dynamics by including a much more complex description and analysis of subsystems and their interactions. In a nutshell, the result of such an effort would be a 'Theory of Everything on Earth'. For example, in the conclusion they envisage the possibility to further enhance the complexity of their approach in including a separate socio-epistemic taxon.

This reviewer is not an expert in the field of modelling. Generally, I think that the paper presents a good overview and offers a glimpse at how such type of models might look like, especially in their toy model. Yet, in my view there are several fundamental problems (what I am writing is against the background of my own work in the philosophy of ecological economics, especially https://doi.org/10.1016/j.ecolecon.2018.03.024 and https://doi.org/10.1016/j.ecolecon.2019.106526) . In addition, I commented extensively on an earlier working paper version, and my impression is that some of my points have not been adequately taken into account in this new version, but it seems to me that this is not a question of right or wrong, but a question, well, of different disciplinary worldviews).

The first point is what kind of data and methodologies are considered as 'state of the art' in modelling. The authors clearly recognize that their approach implies the inclusion of a wide range of other disciplines, even the humanities. But these disciplines do not live in peaceful co-existence: Economists and sociologists often sharply criticize each other, and both may consider the humanities more art than science. What are the implications for integrating widely diverging disciplinary approaches in one integrative modelling approach? My impression is that the authors are aware of the issue, but their reaction seems naïve: Just increase complexity of the model, multiply taxa and so on. Of course, they cannot really tackle this issue in this paper, but one would expect a more sophisticated discussion about the methodological implications. One certainly is that model builders themselves should come from different disciplines, and that a part of the 'modelling' would be a precise method for organizing their collaboration. In other words, the true 'model' would include the people, and not just what happens in their computers and what is manifest in the papers they produce. That is philosophically deeper than it sounds (think of actor-network theory, for example): A model of the type sketched in the paper is a dynamic structure of distributed cognition, and a full

model description must include this meta-level. Indeed, this is obvious from the fact that the authors present a rich and informative overview of existing models across the disciplines, mostly pointing at their limitations. But what follows from that? Plugging the plethora of models together could be done by a lonely genius, or by a diverse team. In that respect, often I notice in the literature a tendency of 'clubbing together', both in co-author and citation clusters. Thinking systematically about the social organization of modelling must be part of the modelling!

That leads me to the second point, with a vengeance: as far as I can see, the authors do not even mention the concept of 'participatory modelling' which is becoming more prominent in ecological economics. This is about the social organization of modelling, again, but beyond science. In that respect, the authors continue to be overly simplistic about their notion of 'World'. They recognize that there is no 'one world', but what is the consequence? In a nutshell, participatory modelling means that the model must make the worlds explicit in which the agents live who drive the systems, and that must be done in asking them to take part in the modelling. This is mainly small-scale and most fitting to ecological modelling, which, as the authors point out, is mostly not done on the scale on which they conduct their modelling. But participatory modelling even further supports this type of 'fragmented modelling', since worlds are specific to groups, and the groups maintain highly various relationships (even open hostility and war). Therefore, I honestly criticize that the authors only pay lip-service to the insight that there is no 'one world'. If there are many worlds, and these are Uexkuellian worlds of ecological and evolutionary subjectivity, how can we include this in the type of models they envision?

The problem of the 'world' has many facets, and my third point is that I am missing a clear ontological grounding of the suggested taxonomy; or, a clear justification of their particular distinction of levels and systems/subsystems. There was a time when 'general systems theory' was fashionable, and one can learn from that. I mentioned that the authors just hide this issue under the slogan of adapting and increasing the complexity of the model. But why exactly didn't they include a socio-epistemic taxon right from the beginning? That's where the disciplinary debates happen! Just think of the anthropological debates about 'cultural materialism' decades ago (Harris versus Sahlins). After all, what are the primary determinants of 'worlds'? The authors cannot avoid tackling such foundational issues: Is it the ideas, or is it the underlying economics, or what else? This shows that there is a deep property of their models: The model is endogenous to what it describes, apparently from an external standpoint. But once

you create a model of that scope and reach, there is no more any external standpoint. That means, if the model explains how, say, ENV and CUL interact, this applies reflexively on their model, since it is part of CULT (the authors mention this!). This is the deeper reason why you need participatory modelling: This breaks the circuit of reflexivity. The scientists will always get stuck in closed loops (just look at the world of economics!).

One aspect of this is providing a foundation for the taxonomy. What could be the alternatives? For example, there are no 'individuals' in the taxonomy. It's all about the larger systems, society, economy, the biophysical systems, and so on. But one could argue that 'agency' should become a taxon of its own. Why? Because the authors aim about 'causal explanations'. That's a big challenge! Can we really build theories of agency on causal explanations? Wouldn't we need a crucial part of the model which is not building on causal explanations, but on theories of agency? In my own work, I have therefore turned to a completely different explanatory framework, semiotics: The stuff of worlds are signs. That would result into a different taxonomy of models. Another example: the entire approach seems heavily anthropocentric. In debates about the notion of Anthropocene, this has been questioned by many scholars. In two of the main taxa we have 'socio'. But there are also good reasons to approach the technosphere as an autonomous domain, analogous to the biosphere. That would also reshuffle cross-disciplinary relations. I believe that the authors should discuss in a more principled way how we can develop scientific standards of taxonomy. Otherwise, what happens is what we see I the paper: They conflate the issue of taxonomy of models with the ontology of systems in their object domain.

Let me end with a personal note. I am deeply skeptical of this type of 'grand modelling', but that is no justification for rejecting the author's approach. The reason is that I work extensively on the relationship between economics and the sciences, not only in the field of ecological economics, but also in the field of neuroeconomics. Neuroscientists rarely build 'grand models' of the brain, it's just too complex. Philosophers of science have developed the so-called 'mechanism approach' or 'constitutive explanations' approach based on this research practice: Neuroscientists study mechanisms, not grand systems. The mechanism approach makes much sense in the social sciences, too, where it is increasingly received. If we regard the brain as too complex to be described in terms of a grand system, why should we hope that this is possible for an even more complex system (if only because it includes brains as parts).

I think that modelers must have the guts to discard claims of integrated 'grand' modelling and go for a fragmented, open-ended and incomplete framework of loosely connected, but empirically well-grounded mechanisms, both in the sense of generalized mechanisms and mechanisms operating at specific times and places. Why 'have the guts'? They are under huge pressure to present forecasts and evaluations to policy makers, and they must at least present the impression that there is 'progress' in research, eventually enabling us to control our 'World-Earth'. That is especially true for those researchers who are really worried about the state of the world. They must make big claims and must suggest that in principle, we could control 'the system', if only we accept our responsibility and our moral duties. Just acknowledging that the world is to messy for us to ever catch it with a model, would be self-defeating, and probably even on moral grounds, if that would imply that competing 'fake news' would reign the world. So, let it be.

---

## Author Response (AR2)

**Taxonomy paper by Donges et al., for Earth System Dynamics, 2nd revision, response letter**

**Editor report**

*Dear authors,*

*based on the completion of the latest round of independent peer review I am very pleased to inform you that I would like to accept your manuscript for publication – subject to minor revisions. After you have made these revisions I will review them. There will be no further requirement for independent peer review.*

*Review 1 report contains some very valuable observations and in places recommendations. I do not think there are any specific issues that need to be addressed as the reviewer says most of these are largely foundational. I would recommend considering their comments about participatory modelling as this may be a good and specific issue with which to at least acknowledge some of the reviewer's concern. I do not think any substantive changes or additions are required here.*

*Reviewer 2 report contains some specific and easy to address issues. I reproduce them here: [...]*

*Once you have made your revisions and upload, I will do my best to ensure a speedy route to publication.*

*Best wishes*
*James Dyke*

Dear Editor,
See below our specific responses to the reviewers.

**Reviewer 1** (Carsten Herrmann-Pillath)

*The World Is Not Enough! On the Limits of 'Grand Modelling'*
*Carsten Herrmann-Pillath*

*The authors present a laudable effort to enrich existing models of Earth system dynamics by including a much more complex description and analysis of subsystems and their interactions. In a nutshell, the result of such an effort would be a 'Theory of Everything on Earth'. For example, in the conclusion they envisage the possibility to further enhance the complexity of their approach in including a separate socio-epistemic taxon.*

*This reviewer is not an expert in the field of modelling. Generally, I think that the paper presents a good overview and offers a glimpse at how such type of models might look like, especially in their toy model. Yet, in my view there are several fundamental problems (what I am writing is against the background of my own work in the philosophy of ecological economics, especially https://doi.org/10.1016/j.ecolecon.2018.03.024 and https://doi.org/10.1016/j.ecolecon.2019.106526). In addition, I commented extensively on an earlier working paper version, and my impression is that some of my points have not been adequately taken into account in this new version, but it seems to me that this is not a question of right or wrong, but a question, well, of different disciplinary worldviews).*

We emphasise again that our taxonomy is diagnostic, not prescriptive! It provides categories and principles to clarify and structure discussions about model combinations in expanded (whether actual, proposed or imagined) Earth system models. The reviewer imbues this with the idea that the end objective of modelling is a unified ontology or 'theory of everything', where all the boxes should be filled. It is precisely that kind of Frankensteinian mess that we want to help avoid, through enabling a more systematic discussion that goes beyond techno-methodological aspects of model combinations.

We welcome the reviewer's suggestion of bringing a deeper philosophical approach to bear on this challenge. In Section 2.4 of our revised text we have now referred to the articles on the function and information aspects of the technosphere (which in the taxonomy mostly spans the MET and CUL taxa) and the philosophical grounding of ecological economics, where arguably more attention to CUL aspects would advance the co-creation of the field.

On p10, a reference has been added to line 6, and text added in lines 15-18: "… in general terms, their view of society contains aspects of our MET taxon, while "the economy" is

more restricted than MET. Hermann-Pillath (2020) argues that the field of ecological economics would benefit from more attention to the creative processes of 'art', which we would frame as CUL aspects that are largely absent from current conceptualisations (as also argued by …)."

> *The first point is what kind of data and methodologies are considered as 'state of the art' in modelling. The authors clearly recognize that their approach implies the inclusion of a wide range of other disciplines, even the humanities. But these disciplines do not live in peaceful co-existence: Economists and sociologists often sharply criticize each other, and both may consider the humanities more art than science. What are the implications for integrating widely diverging disciplinary approaches in one integrative modelling approach? My impression is that the authors are aware of the issue, but their reaction seems naïve: Just increase complexity of the model, multiply taxa and so on. Of course, they cannot really tackle this issue in this paper, but one would expect a more sophisticated discussion about the methodological implications. One certainly is that model builders themselves should come from different disciplines, and that a part of the 'modelling' would be a precise method for organizing their collaboration. In other words, the true 'model' would include the people, and not just what happens in their computers and what is manifest in the papers they produce. That is philosophically deeper than it sounds (think of actor-network theory, for example): A model of the type sketched in the paper is a dynamic structure of distributed cognition, and a full model description must include this meta-level. Indeed, this is obvious from the fact that the authors present a rich and informative overview of existing models across the disciplines, mostly pointing at their limitations. But what follows from that?*

Here too, our intention is much simpler than the reviewer suggests in the question above. We are not promoting efforts in the direction of "one integrative modelling approach"; we are observing that Earth system analysis already is integrating models. Model builders may come from different disciplines – and this may be desirable for many reasons, not least relating to the social realities of academic life and the place of science in society – but we would not argue that they necessarily "should" do so (eg, economic modellers don't work with physicists despite the physics origins of key modelling concepts.) When insights, representation techniques, algorithms etc are taken from multiple fields of knowledge, it is helpful for model-makers, model users and model critics alike to be able to categorise those domains not by discipline but by the kinds of interactions. Our taxonomy is not a structure for a model; it is a structure for enabling meta-analytic (if not necessarily metatheoretical) conversation about models and modelling.

We now make this clearer:

- in added text in the abstract ("combine and critique model components…");
- on p2: replacing "the World-Earth system" with "World-Earth systems", to indicate we envisage the same kind of diversity and pluralism in these new models as in current Earth system models and integrated assessment models;
- also on p2, adding clauses "should now be included on equal terms in a new family of models to conduct systematic global analyses of the Anthropocene"; and "nor do we intend it to serve as a universal blueprint for models of essentially everything".
- on p10 line 24-25 (also in response to reviewer 2), adding "The taxonomy approach means that things that were previously included in models as opaque and unquestioned systems can be unpacked and critically examined. This would be of particular benefit to model users who were not the model builders."
- adding reference in the conclusions to earlier discussions by Schellnhuber and Steffen et al about how World-Earth systems understanding and modelling efforts can evolve
- adding a sentence to the concluding remarks: "By supporting the development and discussion of new family of models, and not by pushing for a rigid and universalising model of everything, the taxonomy promises…"

> *Plugging the plethora of models together could be done by a lonely genius, or by a diverse team. In that respect, often I notice in the literature a tendency of 'clubbing together', both in co-author and citation clusters. Thinking systematically about the social organization of modelling must be part of the modelling!*

This would be a fascinating exploration for a future study. The individual contributions and social organisation of modelling have been examined for several other contexts (a prominent example is Nordhaus and DICE). For Earth system analysis, arguably the field has been one of international collaborative strategic design more than a tradition of a lonely genius plugging bits together, making it harder to see this as a tractable approach now.

> *That leads me to the second point, with a vengeance: as far as I can see, the authors do not even mention the concept of 'participatory modelling' which is becoming more prominent in ecological economics. This is about the social organization of modelling, again, but beyond science. In that respect, the authors continue to be overly simplistic about their notion of 'World'. They recognize that there is no 'one world', but what is the consequence? In a nutshell, participatory modelling means that the model must make the worlds explicit in which the agents live who drive the systems, and that must be done in asking them to take part in the modelling. This is mainly small-scale and most fitting to ecological modelling, which, as the authors point out, is mostly not done on the scale on which they conduct their modelling. But participatory modelling even further supports this type*

*of 'fragmented modelling', since worlds are specific to groups, and the groups maintain highly various relationships (even open hostility and war). Therefore, I honestly criticize that the authors only pay lip-service to the insight that there is no 'one world'. If there are many worlds, and these are Uexkuellian worlds of ecological and evolutionary subjectivity, how can we include this in the type of models they envision?*

We tackle the rich themes raised in this paragraph in reverse order. We now refer to "World-Earth systems" in plural, not singular, throughout the paper, so the text now indicates more explicitly that many framings and representations are possible in World-Earth modelling.

Any given model will (likely) represent one framing of the system, just as they do in Earth system modelling. For example, in ESMs, the different representations of atmospheric processes in energy-balance models and general circulation models enable or constrain coupling of different subsystems – ocean, land, cryosphere, etc. Thus choices about representations in the different taxa will have consequences for the design and viable combinations of components of World-Earth models. A purely subjective 'world' might not be one of the many worlds that can be integrated in World-Earth models, for instance if (say) it obviates the analytic or predictive power conferred by 'objective' things such as the specific heat capacity and freezing point of water.

We disagree that all world-modelling must ask the agents to take part in the modelling. We view participatory modelling as an example of how a system can come to be represented in a model, not what is represented (although in some applications the participation becomes the model, such as in real games). However, our taxonomy might be useful in participatory processes, so we add the following to our conclusions:

p21 lines 8-10: "It can help with operational model development as is illustrated by the work reported in the companion paper (Donges et al. 2020). It can also help in interdisciplinary communication, model critique, and potentially even participatory modelling processes by providing an organisational scheme and a shared vocabulary to refer to the different components that need to be brought together."

*The problem of the 'world' has many facets, and my third point is that I am missing a clear ontological grounding of the suggested taxonomy; or, a clear justification of their particular distinction of levels and systems/subsystems. There was a time when 'general systems theory' was fashionable, and one can learn from that. I mentioned that the authors just hide this issue under the slogan of adapting and increasing the complexity of the model.*

*But why exactly didn't they include a socio-epistemic taxon right from the beginning? That's where the disciplinary debates happen! Just think of the anthropological debates about 'cultural materialism' decades ago (Harris versus Sahlins). After all, what are the primary determinants of 'worlds'? The authors cannot avoid tackling such foundational issues: Is it the ideas, or is it the underlying economics, or what else? This shows that there is a deep property of their models: The model is endogenous to what it describes, apparently from an external standpoint. But once you create a model of that scope and reach, there is no more any external standpoint. That means, if the model explains how, say, ENV and CUL interact, this applies reflexively on their model, since it is part of CULT (the authors mention this!). This is the deeper reason why you need participatory modelling: This breaks the circuit of reflexivity. The scientists will always get stuck in closed loops (just look at the world of economics!).*

*One aspect of this is providing a foundation for the taxonomy. What could be the alternatives? For example, there are no 'individuals' in the taxonomy. It's all about the larger systems, society, economy, the biophysical systems, and so on. But one could argue that 'agency' should become a taxon of its own. Why? Because the authors aim about 'causal explanations'. That's a big challenge! Can we really build theories of agency on causal explanations? Wouldn't we need a crucial part of the model which is not building on causal explanations, but on theories of agency? In my own work, I have therefore turned to a completely different explanatory framework, semiotics: The stuff of worlds are signs. That would result into a different taxonomy of models. Another example: the entire approach seems heavily anthropocentric. In debates about the notion of Anthropocene, this has been questioned by many scholars. In two of the main taxa we have 'socio'. But there are also good reasons to approach the technosphere as an autonomous domain, analogous to the biosphere. That would also reshuffle cross-disciplinary relations. I believe that the authors should discuss in a more principled way how we can develop scientific standards of taxonomy. Otherwise, what happens is what we see in the paper: They conflate the issue of taxonomy of models with the ontology of systems in their object domain.*

We reiterate that the taxonomy is NOT a model blueprint. Of course there are ontological aspects to this taxonomy – and we deliberately do not want to narrow their possibilities, just make them more readily visible. For instance, Figure 1 helps to show that we do not view ENV and CUL as necessarily incommensurate, despite their very different ontological foundations.

To make this more concrete: "secularization" does not figure at all in the materialist positivist ontology of ENV representations of climate and the water cycle, just as the specific heat capacity of water (to return to that earlier example) does not have a place in models of the social construction of the spiritual meaning of sacred springs. Yet an World-Earth system analyst may actually choose to combine these entry points in a model. Such models are currently given the expansive umbrella labels of "hybrid" or "integrated" models, but we argue that it would be useful to be able to describe such an effort more explicitly and systematically as, say, a hybrid CUL→ENV model.

We do not expand on the socio-epistemic sub-taxon idea for the reasons we give when we mention it, namely, we see science & technology as a small part of wider culture, and the taxonomy is focused on a compact set of "higher-level" taxa.

> *Let me end with a personal note. I am deeply skeptical of this type of 'grand modelling', but that is no justification for rejecting the author's approach. The reason is that I work extensively on the relationship between economics and the sciences, not only in the field of ecological economics, but also in the field of neuroeconomics. Neuroscientists rarely build 'grand models' of the brain, it's just too complex. Philosophers of science have developed the so-called 'mechanism approach' or 'constitutive explanations' approach based on this research practice: Neuroscientists study mechanisms, not grand systems. The mechanism approach makes much sense in the social sciences, too, where it is increasingly received. If we regard the brain as too complex to be described in terms of a grand system, why should we hope that this is possible for an even more complex system (if only because it includes brains as parts).*

> *I think that modelers must have the guts to discard claims of integrated 'grand' modelling and go for a fragmented, open-ended and incomplete framework of loosely connected, but empirically well-grounded mechanisms, both in the sense of generalized mechanisms and mechanisms operating at specific times and places. Why 'have the guts'? They are under huge pressure to present forecasts and evaluations to policy makers, and they must at least present the impression that there is 'progress' in research, eventually enabling us to control our 'World-Earth'. That is especially true for those researchers who are really worried about the state of the world. They must make big claims and must suggest that in principle, we could control 'the system', if only we accept our responsibility and our moral duties. Just acknowledging that the world is to messy for us to ever catch it with a model, would be self-defeating, and probably even on moral grounds, if that would imply that competing 'fake news' would reign the world. So, let it be.*

We are actually coming from the same position of skepticism about 'grand modelling' ambitions and claims. The taxonomy provides a way for scholars to expose, structure and discuss the differences between modelling efforts, not a structure to promote this universalizing approach. We hope that it will help people to recognize when and in what ways their model toolkit is incomplete or inconsistent, because it is in those contexts of use and discussion that we have found it useful ourselves.

**Reviewer 2** (Birgit Müller)

> *End of page 19 on the version with track changes: "Overall the DISCOUNT model…". It seems that sentence is not fully written out (see "…"). Please check.*

Thank you for noticing this mistake. We have completed the point we were trying to make! The revised text now states: "Overall, the DISCOUNT model provides a first test of the taxonomy's guiding principles. It demonstrates the taxonomy's operative capacity to trace links between established dynamical systems methodology and macro behaviour; it is compatible with diverse research fields, here linking carbon cycles and social learning; and it has appropriate compactness, since tracing the loops and flows between taxa in this World-Earth model do not make us need to rethink the whole structure of the taxonomy."

> *Suggestions to the authors:*
> *2. With respect to an answer of the authors to reviewer 1: "The taxonomy approach means that things that were previously included in models as opaque and unquestioned systems can be unpacked and critically examined. Model users who were not the model builders would really benefit from knowing (to take the example above) whether the representation of an education decision process in a model is in MET or CUL."*

> *Suggestion: Perhaps this illustrative short example on education could be added to the manuscript. It points out the additional value of clarifying the assignment of submodule to a certain taxa by the modeller.*

We have added this point as an additional paragraph on page 10, section 2.4:
"The taxonomy approach means that things that were previously included in models as opaque and unquestioned systems can be unpacked and critically examined. This would be of particular benefit to model users who were not the model builders. For example, education may be explicitly linked to demography (as in various integrated assessment models), so typically would be treated as a quantifiable and accumulable process in the MET taxon: i.e., investment in women's education results in a lower birth rate and

therefore less future land use. In CUL, education would perhaps be treated in a more relational way dealing with the spread of ideas, development of communities, changes in power structures etc."

> *3. With respect to an answer of the authors to reviewer 2: "Since to our knowledge it is the first published model which endogenizes the choice of discount factors used in climate policy."*
>
> *Perhaps this is also worth to mention as novelty in the paper?*

Thank you for pointing this out. We agree and have emphasised this important point more in the abstract, introduction, model description and conclusion sections:

- p2 lines 1-2 "As an example, we apply the taxonomy to a stylised World-Earth system model that endogenises socially transmitted choice of discount rates describing how much societies value the present relative to the future) in a greenhouse gas emissions game."

- p3 lines 24-25 "...we study an example of a novel World-Earth model that seeks to overcome the long-standing challenge of endogenising the choice of discount factors in climate mitigation studies."

- p17 lines 5-7 "The novelty of this model is that it endogenises socially transmitted choice of discount rates in a greenhouse gas emissions game to illustrate the effects of social-ecological feedback loops that are so far typically not considered in current climate economics and IAM modelling efforts."

- p22 lines 3-5 "We use the Copan:DISCOUNT model to demonstrate the value of the taxonomy for tracing how dynamics and feedbacks loop through different taxa, enabling better model design and communication about path-breaking approaches to World-Earth modelling."